# Epitope resurfacing on dengue virus-like particle vaccine preparation to induce broad neutralizing antibody

Wen-Fan Shen[1†], Jedhan Ucat Galula[2†], Jyung-Hurng Liu[3†], Mei-Ying Liao[2†], Cheng-Hao Huang[2], Yu-Chun Wang[2], Han-Chung Wu[4], Jian-Jong Liang[5], Yi-Ling Lin[5], Matthew T Whitney[6], Gwong-Jen J Chang[6], Sheng-Ren Chen[7], Shang-Rung Wu[7,8*], Day-Yu Chao[2†*]

[1]Microbial Genomics Ph.D. Program, National Chung Hsing University and Academia Sinica, Taichung City, Taiwan; [2]Graduate Institute of Microbiology and Public Health, College of Veterinary Medicine, National Chung-Hsing University, Taichung City, Taiwan; [3]Institute of Genomics and Bioinformatics, College of Life Science, National Chung-Hsing University, Taichung City, Taiwan; [4]Institute of Cellular and Organismic Biology, Academia Sinica, Taipei, Taiwan; [5]Institute of Biomedical Sciences, Academia Sinica, Taipei, Taiwan; [6]Division of Vector-Borne Diseases, Centers for Disease Control and Prevention, Fort Collins, Colorado, United States; [7]Institute of Oral Medicine, College of Medicine, National Cheng Kung University, Tainan, Taiwan; [8]Institute of Basic Medical Sciences, College of Medicine, National Cheng Kung University, Tainan, Taiwan

*For correspondence:
shangrungwu@gmail.com (SW);
dychao@nchu.edu.tw (DC)

†These authors contributed equally to this work

Competing interests: The authors declare that no competing interests exist.

**Abstract** Dengue fever is caused by four different serotypes of dengue virus (DENV) which is the leading cause of worldwide arboviral diseases in humans. Virus-like particles (VLPs) containing flavivirus prM/E proteins have been demonstrated to be a potential vaccine candidate; however, the structure of dengue VLP is poorly understood. Herein VLP derived from DENV serotype-2 were engineered becoming highly matured (mD2VLP) and showed variable size distribution with diameter of ~31 nm forming the major population under cryo-electron microscopy examination. Furthermore, mD2VLP particles of 31 nm diameter possess a T = 1 icosahedral symmetry with a groove located within the E-protein dimers near the 2-fold vertices that exposed highly overlapping, cryptic neutralizing epitopes. Mice vaccinated with mD2VLP generated higher cross-reactive (CR) neutralization antibodies (NtAbs) and were fully protected against all 4 serotypes of DENV. Our results highlight the potential of 'epitope-resurfaced' mature-form D2VLPs in inducing quaternary structure-recognizing broad CR NtAbs to guide future dengue vaccine design.
DOI: https://doi.org/10.7554/eLife.38970.001

## Introduction

Dengue virus (DENV), a member of the family *Flaviviridae*, is a mosquito-borne pathogen with four distinct serotypes, DENV-1 to DENV-4 (*Lindenbach and Rice, 2001*). It has been estimated that DENV infects about 390 million individuals globally each year, resulting in 96 million clinically apparent infections ranging from mild fever to the life-threatening dengue hemorrhagic fever (DHF) or dengue shock syndrome (DSS) (*Bhatt et al., 2013*). Although the chimeric yellow fever 17D-derived, tetravalent dengue vaccine (CYD-TDV) has recently been approved by the governments of a few DENV-circulating countries, the unexpected low vaccine efficacy in dengue-naïve children or children younger than 6 years old has limited the use of this vaccine in the 9 – 45 age group living in endemic

countries (*Guy and Jackson, 2016*). Developing a new strategy by either improving the efficacy of CYD-TDV (*WHO, 2017*; *Aguiar et al., 2016*; *Halstead, 2017*) or the second-generation vaccine candidates (*Kirkpatrick et al., 2016*) is essential to broaden the coverage for all vulnerable populations.

The envelope (E) protein of DENV on the surface of viral particles is the major target of neutralizing antibodies (NtAbs) (*Roehrig, 2003*; *Pierson et al., 2008*). The ectodomain of the E protein contains three distinct domains, EDI, EDII, and EDIII, which are connected by flexible hinges to allow rearrangement of domains during virus assembly, maturation and infection (*Modis et al., 2003*). However, the immune response from humans who have recovered from primary DENV infections is dominated by cross-reactive (CR), non-NtAbs which recognize mainly pre-membrane (prM) or fusion loop of the E (FLE) protein (*Dejnirattisai et al., 2010*). During virus replication, newly synthesized DENVs are assembled as immature particles in the endoplasmic reticulum at neutral pH, followed by translocation through the trans-Golgi network and low-pH secretory vesicles (*Chambers et al., 1990*). The pr portion of the prM protein, positioned to cover the fusion loop (FL) peptide at the distal end of each E protein, prevents premature fusion during the virion maturation process along the egress pathway from the infected cells (*Li et al., 2008*; *Kostyuchenko et al., 2013*). For DENV to become fully infectious, the pr molecule has to be removed by a cellular furin protease during the egress process (*Rodenhuis-Zybert et al., 2010*). However, furin cleavage in the low-pH secretory vesicles is thought to be inefficient; hence, DENV particles released from infected cells are heterogeneous populations with different degrees of cleavage and release of the pr portion of prM to form the mature membrane (M) protein (*Pierson and Diamond, 2012*). Immature DENV (imDENV) particles consist of uncleaved prM protein and partially immature particles (piDENV) containing both prM and M proteins, while fully mature DENV (mDENV) contains only M protein (*Pierson and Diamond, 2012*; *Junjhon et al., 2010*). Functional analyses have revealed that a completely immature flavivirus lacks the ability to infect cells unless in the presence of anti-prM antibodies through a mechanism called antibody-dependent enhancement (ADE) of infection (*Dejnirattisai et al., 2010*; *Rodenhuis-Zybert et al., 2010*; *da Silva Voorham et al., 2012*; *Rodenhuis-Zybert et al., 2011*). ADE plays an important role in dengue pathogenesis, and is potentially modulated by the antibody concentration and the degree of virion maturity (*Nelson et al., 2008*; *Rodenhuis-Zybert et al., 2015*).

Virus-like particles (VLPs) containing flavivirus prM/E proteins have been demonstrated to be a potential vaccine candidate, since their ordered E structures are similar to those on the virion surface and also undergo low-pH-induced rearrangements and membrane fusion similar to viral particles (*Allison et al., 1995*). Also, VLP vaccines present several advantages since they are highly immunogenic, non-infectious, and accessible to quality control as well as increased production capacity. However, we theorized that the process of VLP maturation might be similar to that of dengue viral particles, with heterogeneous pr-containing structures. In this study, we engineered and characterized the structure of mature DENV-2 VLPs through cryo-EM. The immunological properties of mature VLPs were further compared with its immature counterparts.

## Results

As we theorized, the process of VLP maturation are similar to that of dengue viral particles so that the wild-type particles (wtD2VLP) produced from the cells are partially immature containing both prM and M proteins (*Figure 1—figure supplement 1*). To overcome this problem, we engineered the DENV-2 VLPs as completely mature particles by manipulating the furin cleavage site at the junction of pr and M in a DENV-2 VLP-expressing plasmid (*Chang et al., 2003*). For comparison, we also generated immature VLP (imD2VLPs) by mutating the minimal furin cleavage motif REKR to REST (*Li et al., 2008*), which resulted in completely uncleaved prM as detected by western blot and ELISA (*Figure 1*). Generation of mature DENV-2 VLP (mD2VLP) was much more challenging. First, we performed multiple sequence alignments of the pr/M junction from several flaviviruses, including the distance-related cell fusion agent virus (CFAV), and analyzed their furin cleavage potential using the algorithm PiTou 2.0[24]. DENV-3 pr/M junction had the lowest predicted Pitou score of 6.87, while WNV had the highest score of 15.4 (*Figure 1A*). Secondly, we determined the cleavage efficiency of D2VLP by replacing P1-8 at the pr/M junction site of DENV-2 with sequences from different flaviviruses. Although the Pitou prediction of the CFAV pr/M junction did not result in a high score for cleavage, replacement with P1-8 of CFAV resulted in the most efficient furin cleavage for D2VLP,

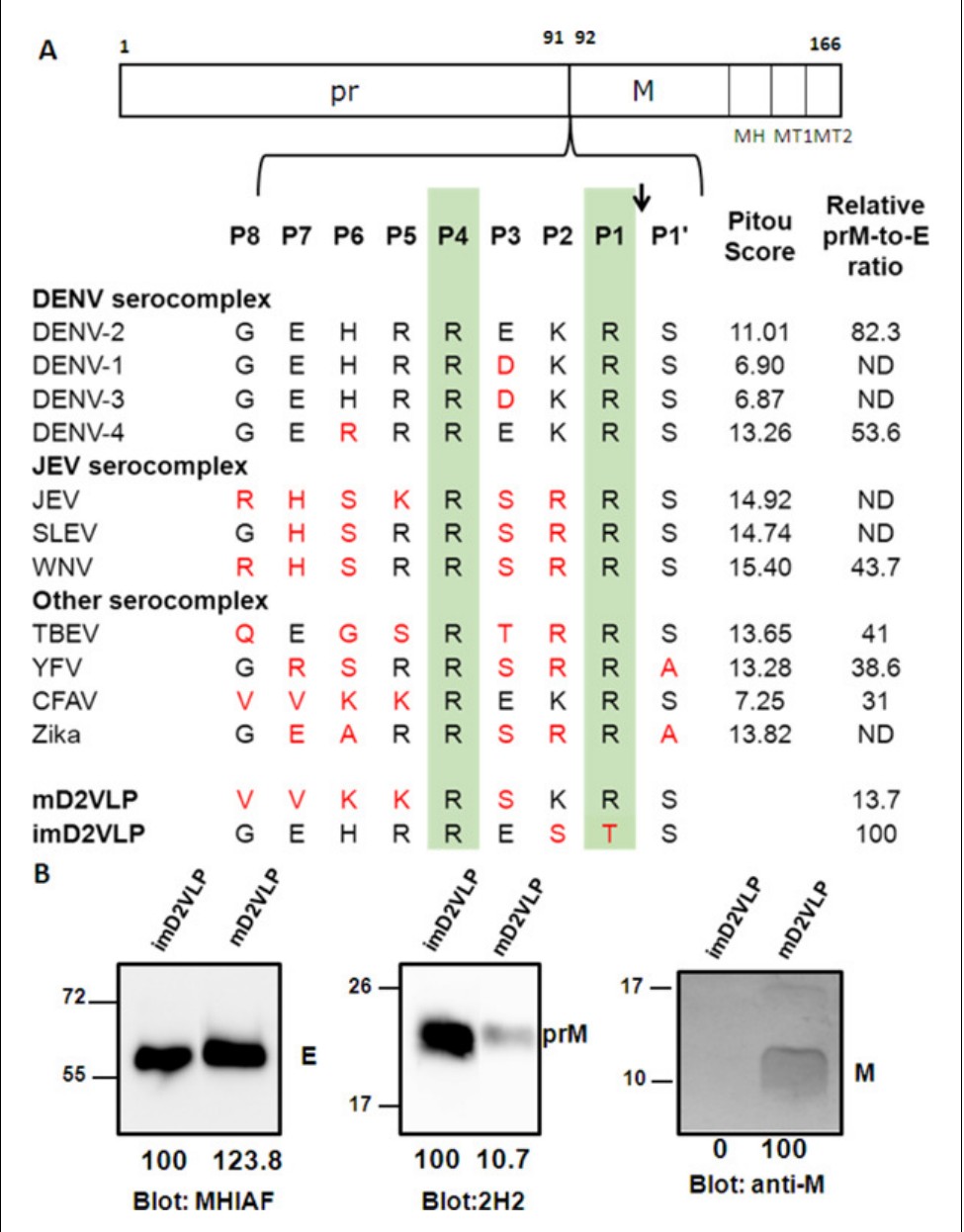

**Figure 1.** Comparison of the prM junction cleavage efficiency among different DENV-2 virus-like particles (D2VLPs). (A) Schematic drawing of the prM protein. The C-terminal of the prM protein contains an α-helical domain (MH) in the stem region, followed by two transmembrane domains (MT1 and MT2). Numbers refer to the position of the amino acids in the polyprotein starting at the first amino acid of prM according to DENV-2 (NP_056776). Single letter designations of amino acid sequence alignment of representative strains from different serocomplexes of flaviviruses at the prM junction site includes dengue virus serotypes 1 – 4 (DENV-1 to DENV-4), Japanese encephalitis virus (JEV), St. Louis encephalitis virus (SLEV), West Nile virus (WNV), tick-borne encephalitis virus (TBEV), yellow fever virus (YFV), cell-fusion agent virus (CFAV), Zika virus (ZKV), immature DENV-2 VLP (imD2VLP) and mature DENV-2 VLP (mD2VLP). Numbers with P in the beginning refer to the positions of the amino acid relative to the prM cleavage site in the proximal direction (without apostrophe) and the distal direction (with apostrophe). Protein sequences were aligned, with the key P1 and P4 positions within the furin cleavage sites highlighted. The arrow indicates the prM cleavage site. The amino acids in red indicate the residues different from DENV-2. The PiTou 2.0 furin cleavage prediction scores are shown on the right for each sequence and the higher scores indicate the higher efficiency of furin cleavage. The relative quantity of prM and E of the wild-type and mutant DENV-2 VLP with P1-8 replacement (from other flaviviruses as shown) was measured by ELISA using MAb 3H5 (specific to E domain III of DENV-2) and MAb 155 – 49 (specific to DENV prM). The relative prM-to-E ratios

*Figure 1 continued on next page*

*Figure 1 continued*

were calculated by absorbance for prM/absorbance for E protein with reference to imD2VLP, whose pr portion was set as 100% uncleaved, as shown on the right. ND: not determined. Data are presented as means from three representative ELISA experiments with two replicates. (B) Culture supernatants of mD2VLP and imD2VLPs were collected and purified after electroporation with the respective plasmids. Five micrograms of proteins were loaded onto a 12% non-reducing Tricine-SDS-PAGE. E, prM and M proteins were assayed by Western blot using mouse hyper-immune ascitic fluids (MHIAF, 1:2000), MAb 2H2 (0.5 μg/mL) and anti-M protein mouse sera (1:25), respectively. E and prM proteins were visualized with enhanced chemiluminesence (ECL); however, M protein was visualized by TMB substrate to avoid high background. The number below each blot shows the relative densitometric quantification of E, prM and M protein bands by Bio-1D software.

DOI: https://doi.org/10.7554/eLife.38970.002

The following figure supplements are available for figure 1:

**Figure supplement 1.** Comparison of prM cleavage among different DENV-2 virus-like particles (D2VLP).
DOI: https://doi.org/10.7554/eLife.38970.003

**Figure supplement 2.** Comparison of physical properties of D2VLP among different DENV-2 virus-like particles (D2VLP).
DOI: https://doi.org/10.7554/eLife.38970.004

---

with only 31% of prM remaining uncleaved as compared to the 100% uncleaved imD2VLP (*Figure 1A*). An additional mutation in the P3 residue from E to S of the CFAV P1-8 construct reduced the prM signaling to 13.7% of that of imD2VLP. We thus named this construct with the replacement of CFAV P1-8 plus the P3 E to S mutation as mD2VLP. The pr/M cleavage of this construct was increased to nearly 90% (*Figure 1B*). Similar to wtD2VLP purified from plasmids transfected culture supernatants, the mD2VLP and imD2VLP showed consistent conformational integrity as ascertained by similar equilibrium banding profiles in 5 – 25% sucrose density gradients, as well as comparable particle size distributions as determined by negative staining electron microscopy, and complex glycosylation on the E proteins (*Figure 1—figure supplement 2*).

Previous study has shown molecular organization of recombinant VLPs from tick-borne encephalitis virus (TBEV) (*Ferlenghi et al., 2001*). However, the cryo-EM results showed low resolution due to particle size heterogeneity and no immunological data can be derived from such structure. To better understand if the structural organization of mD2VLP is similar to that of TBEV and its immunological characteristics, purified mD2VLPs were subjected to cryo-EM analysis. The results demonstrated that the mD2VLPs preparation had a variable size distribution (*Figure 2—figure supplement 1*). In order to perform the single particle 3D reconstruction, first of all, we used a dengue virus-specific Mab (MAb32-6) (*Li et al., 2012*) to identify the mD2VLPs, it turned out that the spherical particles with a diameter of ~31 nm (*Figure 2—figure supplement 2*), which were also the major population as shown in *Figure 2—figure supplement 1B*, were well bound by antibodies. Besides, the particles with diameter of ~31 nm had more solid features in 2D image analyses (*Figure 2—figure supplement 1*) and were able to obtain stable structure potentially than other groups. Therefore, we continued and focused on this population for further 3D reconstruction and image analyses. The results showed that the 31nm-diameter mD2VLPs at a resolution of 13.1 Å had a hollow structure and smooth surface with protrusions around the 5-fold positions (*Figure 2A*, left, and *Figure 2—figure supplement 3*). Fitting the atomic E and M surface proteins PDB: 3J27) into the cryo-EM density map showed that the immunogold labelled mD2VLPs had 60 copies of E packed as dimers in a T = 1 icosahedral surface lattice (*Figure 2A*, right). The apparent differences of structural features on the mD2VLPs compared to the mature native virion particles (*Kostyuchenko et al., 2013*; *Zhang et al., 2013a*) was a slimmer lipid bilayer and the rearrangements of the surface proteins. First, the central section through the viron and m2DVLP reconstructions showed that the bilayer of mD2VLP was relatively thinner than that of viron. The distance between the exterior leaflet and the interior leaflet of mD2VLP and virion was 12 Å and 14 Å, respectively (*Figure 2B*). A similar T = 1 icosahedral symmetry of the E-protein arrangement and thinner lipid bilayer were also found in TBEV VLPs (*Ferlenghi et al., 2001*). Second, interpretation of the map showed that E dimer subunits moved apart from each other causing less density in the intra-dimeric interphase (*Figure 3C*). Because of this loose interaction, a groove located within the E-protein dimer near the icosahedral 2-fold vertices on mD2VLP was noted (*Figure 3A*, right, and *Figure 3—figure supplement 1*). Third,

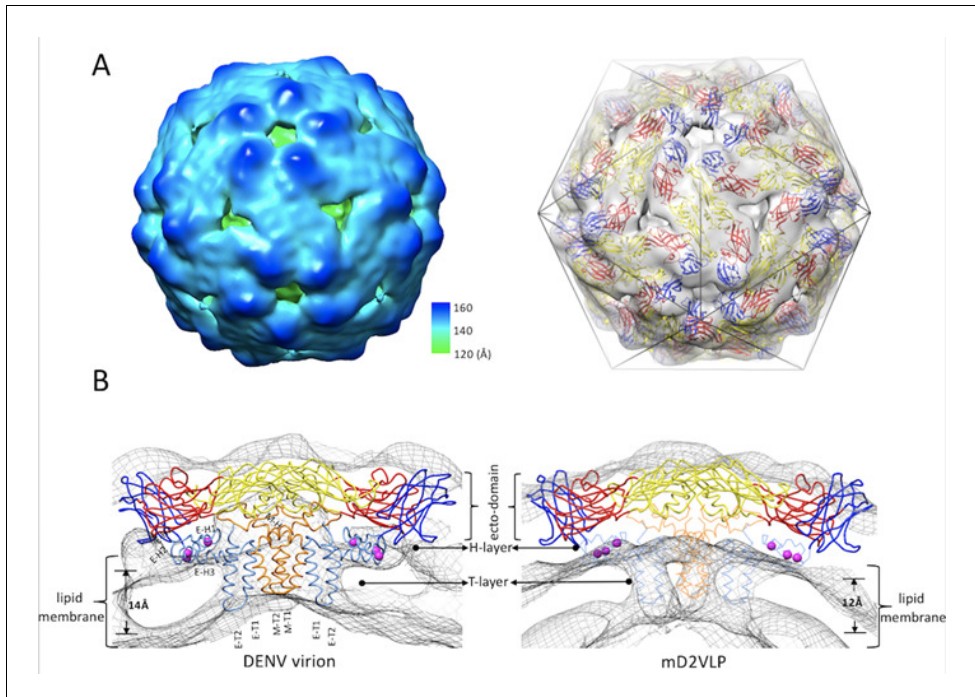

**Figure 2.** The structure of mature form virus-like particles of dengue virus serotype 2 (mD2VLP). (**A**) The reconstructed cryo-EM map of the DENV VLPs (left panel) were presented with the radial color-code indicated (left), the size of the particle is 31 nm. Fitting of atomic E, M surface protein (PDB: 3J27) into cryo-EM density map (right) showed that the VLP has 60 copies of E in a T = 1 arrangement. The density map was shown as a transparent volume rendering into which was fitted the backbone structures of the E ecto-domain. The domains I, II, and III were highlighted in red, yellow and blue, respectively. The cage indicated the icosahedral symmetry. (**B**) The cross section showed the fitted E:M:M:E heterotetramer (PDB: 3J27) into DENV virion (left) and into mD2VLP (right). The map density was in mesh presentation. The atomic model of E:M:M:E heterotetramer was showed in ribbon. Domain definition of dengue E was the same as the previous description, the transmembrane domain of E was colored as light blue, the M protein was in orange color. It was clear that the density of H-layer which is composed of E-H1 to E-H3 and M-H was more solid while the density in T-layer which contains E-T1, E-T2, M-T1 and M-T2 was weak in VLP than in virion. The residues at 398, 401 and 412 in E-H1 of JEV sequence which were proved to play important role in promoting extracellular secretion (*Purdy and Chang, 2005*) were shown as magenta spheres. The transmembrane, perimembrane helices and lipid bilayer were labelled, the critical measurements were also shown.

DOI: https://doi.org/10.7554/eLife.38970.005

The following figure supplements are available for figure 2:

**Figure supplement 1.** The cryo-EM images and 2D analysis of the particles.
DOI: https://doi.org/10.7554/eLife.38970.006
**Figure supplement 2.** The EM images of immunogold-labeled mD2VLPs.
DOI: https://doi.org/10.7554/eLife.38970.007
**Figure supplement 3.** The cryo-EM structure of m2DVLP.
DOI: https://doi.org/10.7554/eLife.38970.008

the central section showed that there was a solid density in the H-layer, which is composed of H helices of E and M protein stem regions, while the density in the T-layer, which contains transmembrane helices of E and M proteins, was weaker in the VLP (*Figure 2B*). Each E protein consists of an E ecto-domain and a stem region that connects the ecto-domain to its transmembrane region. The stem contains two α-helices, one of which interacts with the viral lipid membrane and the other with the E ecto-domain. This looser interaction of E protein dimers was further stabilized by the E protein rearrangement and the closer interaction with one of the α-helices, which resulted in the solid density beneath the ecto-domain as shown in *Figure 2B*. It is worth noting that the stem region of this mD2VLP was replaced into Japanese Encephalitis viral (JEV) sequence; it was previously suggested

that this replacement increased the hydrophobicity for the interaction with the lipid membrane and enhanced the secretion of dengue VLP (*Purdy and Chang, 2005*).

Next, we speculated that the structure of mD2VLP with a groove located within the E-protein dimer could affect the epitope presentation on the surface of the particles. First, we calculated the solvent accessibility of mD2VLP and compared the results with that of DENV-2 virion. This rearrangement of E protein under T = 1 symmetry on the VLP surface exposed more accessible epitopes (48.2%; 191 amino acid with ≥30% solvent accessibility among 396 amino acids of the entire E protein) compared to DENV-2 virions (43.4%; 172 amino acid), particularly located at fusion loop, aa 239, aa 251 – 262 of EDII which together formed the groove, and at A strand, cd loop and G strand of EDIII which surrounded a 5-fold axis (*Figure 3A*, left, *Figure 3—figure supplement 2*). In silico footprint analysis (*Figure 3B*) showed that those cryptic residues on the E protein, which were previously buried inside of the native DENV virion and only interacted with the broad NtAbs while the virion 'breathed' (*Lok, 2016*), were better exposed on the m2DVLP. Specifically, the epitopes interacting with MAb 1A1D-2, which binds to the virus at 37 °C (*Lok et al., 2008*); MAb 2D22, which could block two-thirds of all dimers on the virus surface, depending on the strain (*Fibriansah et al., 2015a*); and the 'E-dimer-dependent epitope', which is recognized by broadly neutralizing MAbs EDE1 and 2 (*Rouvinski et al., 2015*), were all well-exposed in VLPs without steric hindrance. Second, a panel of murine MAb, recognizing different domains of E protein and used previously for antigenic mapping (*Crill et al., 2012*), was subjected to binding-ELISA. The results confirmed that most of the conformational-dependent Mabs preferentially binds to mD2VLP, in particularly those recognizing domain I/II (*Figure 4*).

To further determine the influence of prM cleavage on the immunogenicity of D2VLP, we immunized groups of 4-week-old BALB/c mice with the purified wtD2VLP, imD2VLP or mD2VLP (4 µg/mouse) twice at a 4 week interval. To better represent the neutralizing antibodies at the late convalescent phase, serum samples collected at 8 weeks after the boost were analyzed by antigen-specific ELISA for antibody response against homologous D2VLP antigens with different maturity profiles. In order to precisely quantify the amount of dengue-specific antibodies, we used the same quantity of purified VLPs in the antigen-capture ELISA (*Figure 5A*). WtD2VLP and imD2VLP induced similar ELISA titers of DENV-2-specific IgG antibody against three VLP antigens. However, stronger reactivity against the mature antigen was noted with mD2VLP immunization (1:15,347 vs. 1:6381 vs. 2529 for mD2VLP, imD2VLP and wtD2VLP, respectively) (*Figure 5B*). We also analyzed the neutralizing ability of these sera against the four serotypes of DENV using a 50% antigen focus-reduction micro neutralization test (FRµNT$_{50}$). The mD2VLP immunization group induced a higher and broader neutralizing antibody response against all 4 serotypes of DENV (FRµNT$_{50}$ for DENV-1 = 1:331, DENV-2 = 1:597, DENV-3 = 1:70 and DENV-4 = 1:141), as compared to the imD2VLP vaccinated group (FRµNT$_{50}$ for DENV-1 = 1:100, DENV-2 = 1:207 DENV-3 = 1:64 and DENV-4 = 1:76) (*Figure 5C*).

Since the amino acid sequence is identical except for the mutations at the furin cleavage site, the difference in antibody binding and neutralizing activity between mD2VLP and imD2VLP vaccinated mice sera could result from the difference in induction of CR anti-prM non-NtAbs or anti-E NtAbs recognizing structure-dependent epitopes. To address whether the higher neutralization activity induced by mD2VLP was partly due to the reduction of anti-prM antibodies that are known to be cross-reactive but have no neutralizing activity (*Dejnirattisai et al., 2010*), we performed epitope-blocking ELISAs by using an anti-prM-specific monoclonal antibody (MAb 2H2). MAb 2H2 blocked only 7.59% of the activity of anti-mD2VLP sera but blocked up to 35.26% of antibodies from the imD2VLP-vaccinated groups (*Figure 5—figure supplement 1*). To avoid steric hindrance due to Mab 2H2 binding, we performed site-directed mutagenesis on three amino acids of pr protein based on the following criteria: (1) the conservation of amino acids among all four serotypes; (2) residues interacting with MAb 2H2 (*Wang et al., 2013*); (3) residues not interfering with prM and E interaction (*Li et al., 2008*). As shown in *Figure 5—figure supplement 2*, K26P was a key residue, which significantly decreased the binding of both MAb 2H2 and 155–49 (another cross-reactive murine Mab recognizing pr protein (*Huang et al., 2006*). However, only the mutant Δ2H2-imD2VLP with F1A, K21D and K26P amino acid triple mutations completely prevent the binding of both MAb 2H2 and 155–49 but not that of DENV-2, E-specific Nt-MAb 3H5 (*Figure 5—figure supplement 3*). The sera from immunized mice were tested for their binding ability to the wild-type and mutant-Δ2H2 imD2VLP. The differences in binding of imD2VLP and Δ2H2 imD2VLP were greater for the anti-imD2VLP sera than the anti-mD2VLP sera (*Figure 5D and E*), suggesting that compared to mD2VLP,

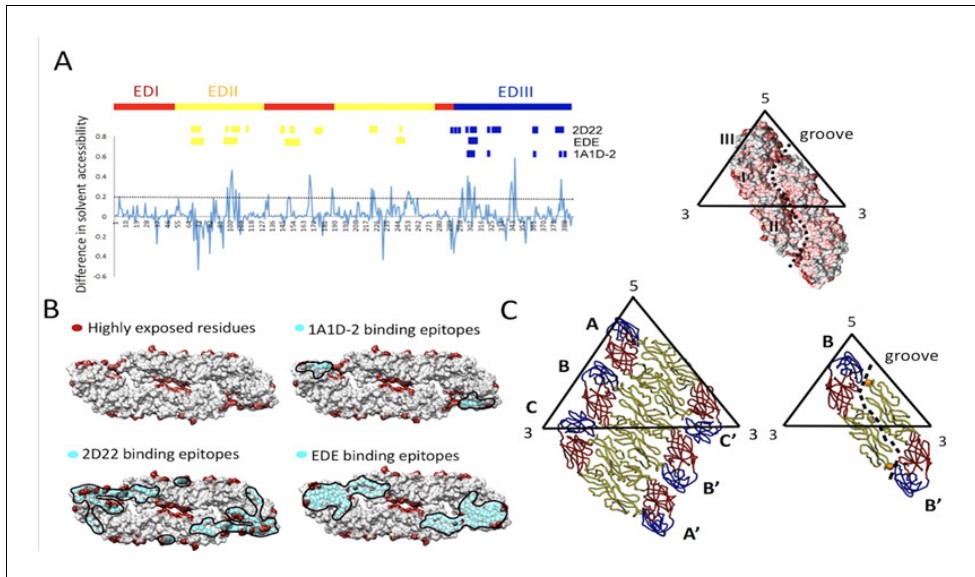

**Figure 3.** Solvent accessibility of dengue virus serotype two virus soluble envelope (sE) protein (A, left) The plot of difference in relative solvent-accessible surface area (Δ%SASA). A positive value of %SASA meant the residue became more exposed when the particle assembly shifts from virion (3 copies of E dimers per icosahedral asymmetric unit) to VLP (1 copy of E dimers per icosahedral asymmetric unit). The black dash line indicated the Δ% SASA $\geq$0.2, which was defined as highly exposed residues in VLP comparing to virion. The high positive values which were focused in the peptide regions such as the fusion loop peptide (including amino acid (aa) residues ranging from 100 to 110), aa 169–170 at domain I, aa 222–226, aa 239 and aa 251–262 at domain II as well as A strand of domain III (aa 300–308), the cd loop of domain III (aa 342–348) and G strand (aa 386–388) around the 5-fold openings. The residues interacting with MAb 1A1D-2[30], including residues 305–312, 352, 364, 388 and 390; the residues interacting with MAb 2D22[31], including residues 67–72, 99, 101–104, 113, 177–180, 225–227, 247, 328, 384–386 (Heavy chain); 148–149, 153–155, 291–293, 295, 298, 299, 307, 309–310, 325, 327, 362–363 (light chain) and the residues interacting with human MAb EDE antibodies (*Rodenhuis-Zybert et al., 2011*), including aa residues 67–74, 97–106, 148–159, 246–249 and 307–314 were indicated. (A, right) The high positive peaks (Δ%SASA $\geq$0.2), low positive peaks (Δ%SASA between 0 and 0.2) and negative peaks (Δ%SASA $\leq$0) in the plot were colored by dark red, deem red and grey in the E dimer surface rendering. The groove located within E-dimer interface was outlined. (B) The highly exposed residues (Δ%SASA >0.2) which were colored by dark red were shown in the surface rendered E-dimer. The residues in E interacting with MAb 1A1D-2, 2D22 and EDE were in cyan spheres showing that they were highly exposed on the m2DVLP surface. Importantly, the binding footprints of the three antibodies were highly overlapping with footprint of highly exposed residues in m2DVLP, and formed a neutralization sensitive patch on m2DVLPs. The areas of the interacting epitopes are circled by black lines. (C) The E protein forming the rafts in virion were shown in the left panel where the three individual E proteins in the asymmetric unit are labeled A, B, and C of the E proteins, in the neighboring asymmetric unit are labeled A', B', and C'. The icosahedral 2-fold E protein dimers (B and B') in m2DVLP have moved apart from each other causing the groove (right). The aa 101 which was responsible for DM25-3 antibody binding were shown in orange spheres.
DOI: https://doi.org/10.7554/eLife.38970.009

The following figure supplements are available for figure 3:

**Figure supplement 1.** The E protein rafts in a virion (left) and mD2VLP (right).
DOI: https://doi.org/10.7554/eLife.38970.010

**Figure supplement 2.** The solvent accessibility analyses.
DOI: https://doi.org/10.7554/eLife.38970.011

**Figure supplement 3.** The 'neutralization-sensitive hotspot' on mD2VLPs.
DOI: https://doi.org/10.7554/eLife.38970.012

imD2VLP was more likely to induce 2H2-like anti-prM antibodies. Thus, mD2VLP has the advantage of eliciting lower levels of anti-prM CR antibody and inducing antibodies with greater neutralizing activity.

The preservation of neutralizing epitopes on the surface of VLPs is critical for efficient production of broadly neutralizing antibody responses. Recent studies have suggested that human monoclonal

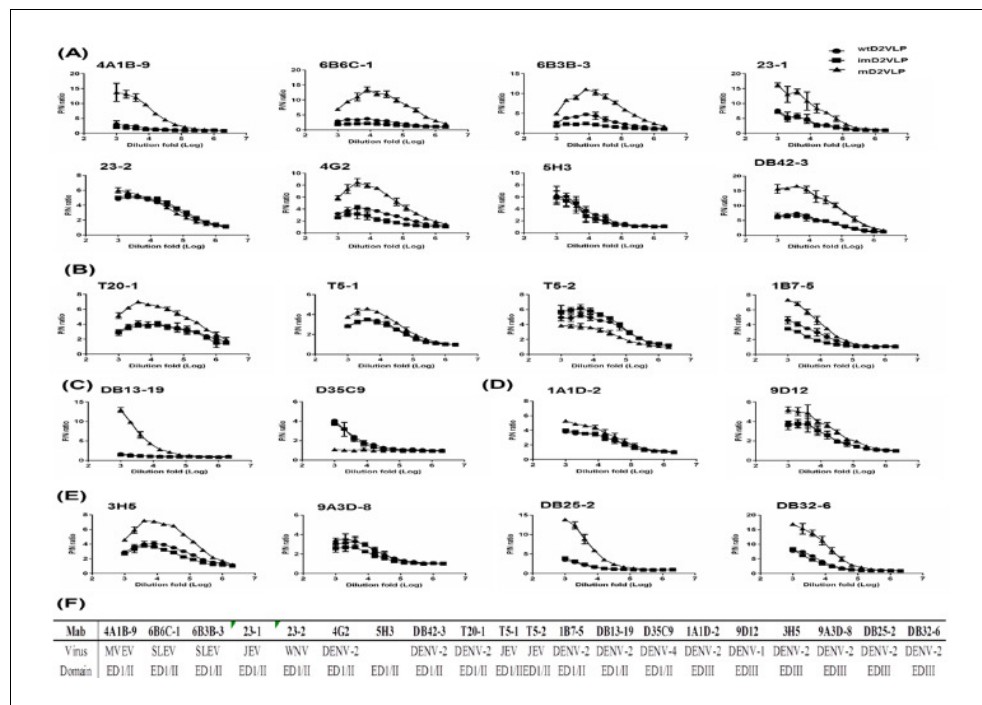

**Figure 4.** Binding avidities of a panel of anti-E monoclonal antibodies (Mabs) to three types of D2VLP with different percentage of pr-peptide on particle surface. Binding curves for (A) eight group-reactive Mabs recognizing E-protein of all four major pathogenic flavivirus serocomplexe, (B) four subgroup-reactive Mabs recognizing more than one flavivirus serocomplex, (C) Two complex-reactive Mabs recognizing all four serotypes of DENV serocomplex viruses, or (D) sub-complex-reactive Mabs recognizing more than one serotypes of DENV, (E) four serotype-specific-reactive Mabs recognizing DENV-2 only, were determined by antigen-capture ELISA. Original immunogen raised for generating murine Mabs and different domain of E protein recognized by Mabs were summarized in (F). Equal quantity of three types D2VLP were first titrated against purified D2VLP before adding to the wells. Mabs were 2-fold serial diluted starting from 1:1000 dilution fold. Data are expressed as P/N value by dividing the OD450 value from each dilution of Mab by the OD450 value from the control COS-1 culture supernatant. Data are means (with standard deviations for binding curves) for duplicates from three representative experiments.

DOI: https://doi.org/10.7554/eLife.38970.013

The following source data is available for figure 4:

**Source data 1.** source data for antibody mapping results in *Figure 4*.

DOI: https://doi.org/10.7554/eLife.38970.014

antibodies (MAbs) reactive with all dengue serotypes can neutralize DENV in the low picomolar range. These MAbs have a preference to bind at the envelope dimer epitopes preserved on virion particles with a high degree of maturity (*Rouvinski et al., 2015*; *Dejnirattisai et al., 2015*). We next investigated if a broad neutralization response to mD2VLP immunization was due to the induction of antibodies with a preference to bind at the E dimer epitopes preserved on the surface of mature VLP (*Rouvinski et al., 2015*; *Dejnirattisai et al., 2015*). The splenocytes from mD2VLP-vaccinated mice were used to perform fusion and generate hybridomas. Among 2836 hybridomas screened, two MAbs with high reactivities to VLP measured by ELISA and FRμNT$_{50}$ were shown in *Figure 6*. As shown in *Figure 6A*, DM8-6 is a serotype-specific MAb reactive only with DENV-2; while MAb DM25-3 recognized all four serotypes of DENV. Next, we tested the neutralizing activity of these two MAbs against four serotypes of DENV. DM8-6 showed good neutralizing activity against DENV-2 at a concentration of 0.037 μg/mL in FRμNT$_{50}$ but poorly neutralized the other three serotypes. Consistent with the ELISA results, DM25-3 neutralized all four serotypes in FRμNT$_{50}$ at 0.32, 0.38, 0.24 and 0.58 μg/mL for DENV-1 to DENV-4, respectively (*Figure 6B*).

To determine if MAb DM25-3 recognized quaternary structure-dependent epitopes presented only on mature virion particles (*Fibriansah et al., 2015a*; *Rouvinski et al., 2015*), DM25-3 was tested

for its binding activity to both mD2VLP and imD2VLP using antigen-capture ELISA. MAb DM25-3 recognized mD2VLP very well, but reacted poorly with imD2VLP (*Figure 6C*). Next, we performed site-directed mutagenesis on the fusion loop peptide in domain II and A-strand in domain III, both of which are important binding regions for CR group/complex neutralizing antibodies (*de Alwis et al., 2011*; *Tsai et al., 2013*) (*Figure 6—figure supplement 1*). The results suggested that amino acid E-101 was likely important in the binding site of DM25-3. Residue 101 on the fusion loop of E protein is conserved among all four serotypes of DENV and can only be exposed when virion particles undergo low-pH induced conformational change or during the 'breathing' state (*Zhang et al., 2015*; *Zhang et al., 2013b*). *Figure 3C* shows that residue 101 is located at the touching point between EDII and EDIII of the dimeric molecule. Since residue 101 is a critical residue in stabilizing E-protein dimers on DENV virion, mutation on 101 would disrupt viral particle formation (*Zhang et al., 2013a*; *Rouvinski et al., 2017*). However, mD2VLP with a mutation of residue 101 can still form particles (*Figure 6—figure supplement 2*), which further support our cryoEM model that the E:E protein interactions on VLP are loose. When the E:E protein interaction was looser, residue 101 at the groove within the dimeric molecule was more exposed (64%) than on the native virion (23.8%, PDB:3J27) or on the native virion during the 37°C 'breathing' state (45.3%, PDB:3ZKO). We also generated a recombinant virus of DENV-2 whose EDIII was exchanged for a consensus EDIII (*Chen et al., 2013*), and the recombinant virus PL046cEDIII was recovered from the transfected cell culture supernatants for use in ELISA and FRμNT. Compared to the parental DENV-2 strain PL046, there was significant loss of binding of MAb DM25-3 and immune mouse sera to PL046cEDIII (*Figure 6D*), suggesting that DM25-3 could be an E-dimer inter-domain antibody whose binding footprint is sensitive to amino acid changes in the EDIII domain. By comparing FRμNT$_{50}$ tests using murine antisera on both PL046 and PL046cEDIII viruses, we found that anti-mD2VLP sera showed a greater difference in neutralizing activity against PL046 and PL046cEDIII than did anti-imD2VLP sera, indicating a greater conformational dependence for anti-mD2VLP-triggered neutralization (*Figure 6E*).

The major obstacle of dengue vaccine development is the lack of a suitable small animal model to evaluate vaccine efficacy. Therefore, we decided to test whether the monovalent mD2VLP immunogen could provide protection from a lethal dose challenge of heterologous dengue virus serotypes using suckling mice developed in a previous study (*Chang et al., 2003*; *Galula et al., 2014*; *Hughes et al., 2012*). The advantage of using sucking mice to test vaccine efficacy is that the protection from dengue virus challenge can only come from maternal antibodies generated by vaccinated female mice and passively transferred to the their babies. The immunization and challenge schedule is illustrated in *Figure 7A*. As shown in *Figure 7B*, suckling mice born from mothers vaccinated with mD2VLP were all protected from challenge with all four serotypes of dengue virus.

## Discussion

How VLP immunogens mimic their viral counterparts structurally and how the neutralizing epitopes are preserved on the VLP surface area of significant interest in the development of good vaccine candidates. Recent studies suggest that potent human neutralizing antibodies with broad reactivity across dengue serocomplex can be generated from dengue patients, particularly after secondary infection (*Rouvinski et al., 2015*; *Dejnirattisai et al., 2015*; *Tsai et al., 2013*). Our unique findings here suggested that mD2VLP, with diameter of 31 nm and a scaffolding of multiple E-protein dimers similar to that of virion, could be capable of inducing such CR antibodies with broad neutralizing activity. This monovalent mature-form dengue VLP with 'epitope re-surfaced' has the potential to be the 'epitope-focused' antigens when combined with other live attenuated dengue vaccine to induce higher and broader neutralizing antibodies (*Rouvinski et al., 2017*; *Rey et al., 2018*).

Very few flavivirus studies explored the immunogenicity of prM-reduced VLP antigens and its capability in inducing broad NtAbs (*Rouvinski et al., 2017*; *Keelapang et al., 2013*; *Suphatrakul et al., 2015*; *Metz et al., 2017*). In TBEV study, VLP preparation showed a range of size distribution with the majority fell into 31 nm (*Ferlenghi et al., 2001*). Greater than 90% of the particles are mature and immunogenic. On the contrary, WNV VLPs were secreted as large (40–50 nm) and small (20–30 nm) particle sizes, which showed different maturity and immunogenicity. The large VLPs with the size of virions (50 nm) are more mature and induced higher NtAb titers and more potent protection against WNV challenge than the small immature VLPs (*Ohtaki et al., 2010*).

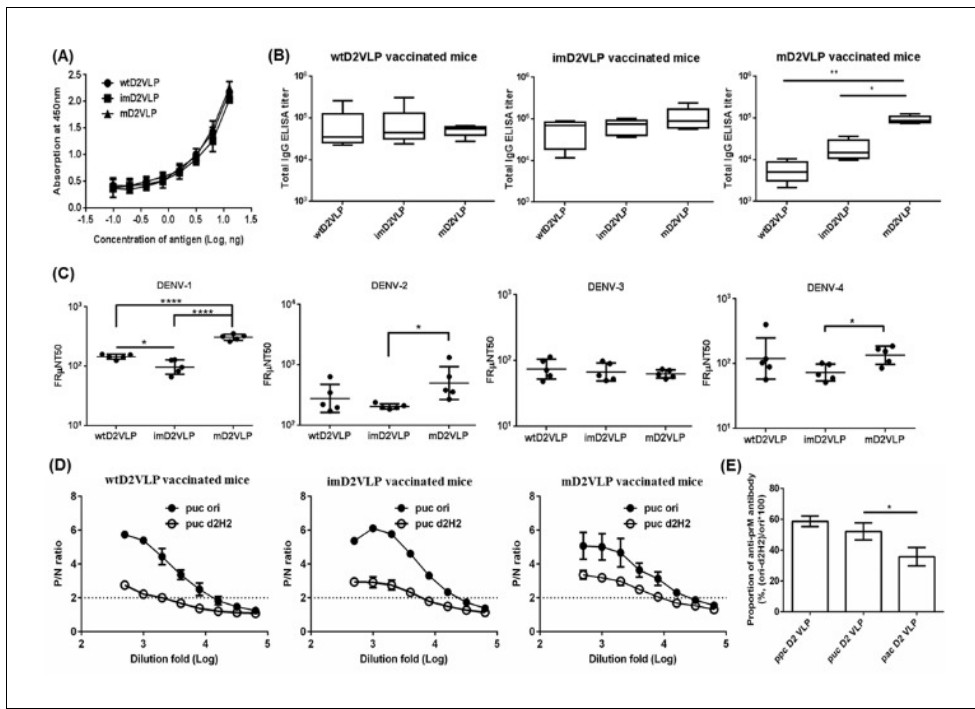

**Figure 5.** Total antigen-specific IgG, neutralizing titers and proportion of anti-prM antibodies compared among three groups of mice immunized with wtD2VLP, imD2VLP or mD2VLP. (A) D2VLPs were concentrated and purified from clarified supernatants. The total protein concentration of purified imD2VLP and mD2VLP were first determined by the Bradford assay and then subjected to antigen-capture ELISA using 2-fold serial dilutions. The standard curve was used to titrate both antigens as equal amounts for the subsequent assays. (B) The endpoint IgG titer of 12 week post-immunization mouse sera was measured by antigen-capture ELISA, using equal amounts of homologous and heterologous purified D2VLP antigens. All endpoint titers were $\log_{10}$ transformed and depicted as geometric means with 95% confidence intervals. (C) The neutralizing antibody titers at 50% antigen focus-reduction micro neutralization (FRμNT50) in Vero cells infected with DENV-1 to 4. (D) Binding reactivity of serial dilutions of anti-wtD2VLP (left), anti-imD2VLP (center), and anti-mD2VLP (right) mouse sera were analyzed by ELISA using equal amounts of wild-type (imD2VLP) and mutant imD2VLP (Δ2H2) antigens. (E) Proportions of 2H2-like, anti-prM antibodies from two different D2VLP immunization groups were calculated based on the formula 100*[(OD450imD2VLP-OD450Δ2H2)/OD450imD2VLP] at a 1:1000 dilution of mouse sera. All data presented are based on a representative of three independent experiments with two replicates from n = 5 mice sera per group per experiment and expressed as mean ±SEM. The statistical significance was determined using the two-tailed Mann-Whitney $U$ test to account for non-normality of the transformed data. *p<0.05; **p<0.01; ****p<0.0001.
DOI: https://doi.org/10.7554/eLife.38970.015

The following source data and figure supplements are available for figure 5:

**Source data 1.** source data for antibody mapping results in *Figure 5*.
DOI: https://doi.org/10.7554/eLife.38970.020

**Figure supplement 1.** Proportions of anti-prM antibodies from both D2VLP immunization groups were measured using an epitope-blocking ELISA.
DOI: https://doi.org/10.7554/eLife.38970.016

**Figure supplement 2.** Epitope identification of MAbs 2H2 and 155–49.
DOI: https://doi.org/10.7554/eLife.38970.017

**Figure supplement 3.** Serial dilutions of imD2VLP and mutant imD2VLP (Δ2H2), containing mutations at the MAb 2H2 binding site (F1A, K21D, K26P), were tested for binding with MAb 2H2 (left) and control antibody 3H5 (right) by ELISA.
DOI: https://doi.org/10.7554/eLife.38970.018

**Figure supplement 4.** Binding ELISA was performed to test the reactivity of sera from mice immunized with two doses of imD2VLP or mD2VLP against equal amounts of imD2VLP and mD2VLP antigens.
DOI: https://doi.org/10.7554/eLife.38970.019

Similar design of prM-reduced D2VLP was used as immunogens but was produced from mosquito cells and such VLP required adjuvants to boost immunity (*Suphatrakul et al., 2015*). Other attempts using E-dimers without the co-expression of prM cannot only display the quaternary structure epitopes but also reduce the exposure of FLE epitopes (*Rouvinski et al., 2017*; *Metz et al., 2017*; *Metz et al., 2018*). However, the immunogenicity of such E-dimers with only two copies of E could be lower than VLP. The mD2VLP used in this study was produced from mammalian cells with the features similar to dengue virion, such as glycosylation pattern, particle distribution in gradient after high-speed centrifugation, epitopes preserved on the surface by the mapping of monoclonal antibodies, and mostly importantly, multiple copies of E structurally packed as dimers on the particle surface lattice (*Figure 1—figure supplement 2* and *Figure 5—figure supplement 1*). Compared to imD2VLP (35%), only 7.6% of prM-recognizing antibodies were induced by mD2VLP through 2H2-blocking assay. Current dengue vaccine candidates in the market such as CYD-TDV or others in clinical trial are either prM-possessing forms or E-protein monomer which is lack of quaternary epitopes (*Wichmann et al., 2017*). Our study suggested that mD2VLP has the potential to be a safer immunogen as the second-generation dengue vaccine.

In addition, mD2VLP-vaccinated mice produce antisera predominantly composed of about 48% of DM25-3-like antibodies (*Figure 5—figure supplement 4*). The packing of E dimers on VLPs does not just preserve quaternary structure epitopes on the surface lattice as suggested previously (*Metz et al., 2018*; *Crill et al., 2009*), but also provides a unique surface-accessible structure with increased epitope accessibility. Usually these epitopes are cryptic in mature virions maintained at 28°C and in a neutral-pH environment (*Lok, 2016*). By superimposing the E-dimer-dependent quaternary epitopes (*Lok et al., 2008*; *Fibriansah et al., 2015a*; *Rouvinski et al., 2015*) onto the structure of mD2VLP, the binding footprints of these antibodies were highly overlapping and formed a 'neutralization-sensitive hotspot' on mD2VLP (*Figure 3—figure supplement 3*). In spite of its size heterogeneity, the underlying mechanisms of why mD2VLPs are able to stimulate an elevated and broader immune response could be based on the following: (1) epitope accessibility exposed at the grooves within E-protein dimers, which govern the generation of neutralizing antibodies; (2) removal of decoy epitopes presented on prM-containing structures found in imD2VLPs; and (3) inter-dimeric epitope accessibility due to the 5-fold and 3-fold openings of the E protein arrangement of T = 1. Other mechanisms cannot be excluded such as reducing the production of FLE-like antibodies (ie. MAb E53 or 4G2), which have a preference of binding to spikes on noninfectious, immature flavivirions by engaging the highly conserved fusion loop that has limited solvent exposure of the epitope on mature virions (*Cherrier et al., 2009*).

The current obstacle in developing the second generation dengue vaccine is that dengue vaccine-induced neutralizing antibodies failed to correlate with or predict vaccine mediated protection (*Yang et al., 2018*; *Moodie et al., 2018*). The discrepancy between neutralization titer and protection against all four serotypes of DENV can also be found in our study (*Figure 5C* and *Figure 7B*). Previous studies have shown that neutralizing activity can be modulated by the maturity of virion in flavivirus (*Pierson et al., 2008*; *Nelson et al., 2008*). Several murine cross-reactive weak neutralizing antibodies, such as 4G2, can have neutralizing activity against all four serotypes of DENV at high concentrations, although they can enhance the virus replication while at lower concentrations (*Zellweger et al., 2010*). To confirm this argument, we mutated amino acid 101 on fusion peptide of imD2VLP and compared the loss of binding of antibodies from the immuned mice sera with the wild-type imD2VLP. Mutation of amino acid 101 on imD2VLP disrupted the binding of murine Mab 4G2. Meanwhile, the results showed that the proportion of 4G2-like antibodies among the sera receiving imD2VLP immunization was greater than that of sera either receiving mD2VLP or wtD2VLP (*Figure 6—figure supplement 3*). Recent studies suggested that the most potent neutralizing antibodies came from those recognizing quaternary epitopes on the smooth surface of dengue virions (*Rouvinski et al., 2015*; *Fibriansah et al., 2015b*; *Crowe, 2017*). Our study suggested that different types of D2VLP could raise different proportions of NtAbs, which are highly structure-dependent and could sometimes contribute to similar level of FRNT50, such as against DENV-3 (*Figure 5C*). Despite the size heterogeneity, we have found that a mature form of monovalent VLP from dengue virus serotype two with 'epitope re-surfaced' is efficient in inducing elevated and broadly NtAbs targeting quaternary epitopes. We cannot exclude the possibility of that the protection of mD2VLP immunization against all four serotypes of DENV challenge could also be due to non-neutralizing mechanism such as antibody-dependent cell-mediated cytotoxicity (ADCC). Therefore, it is crucial to

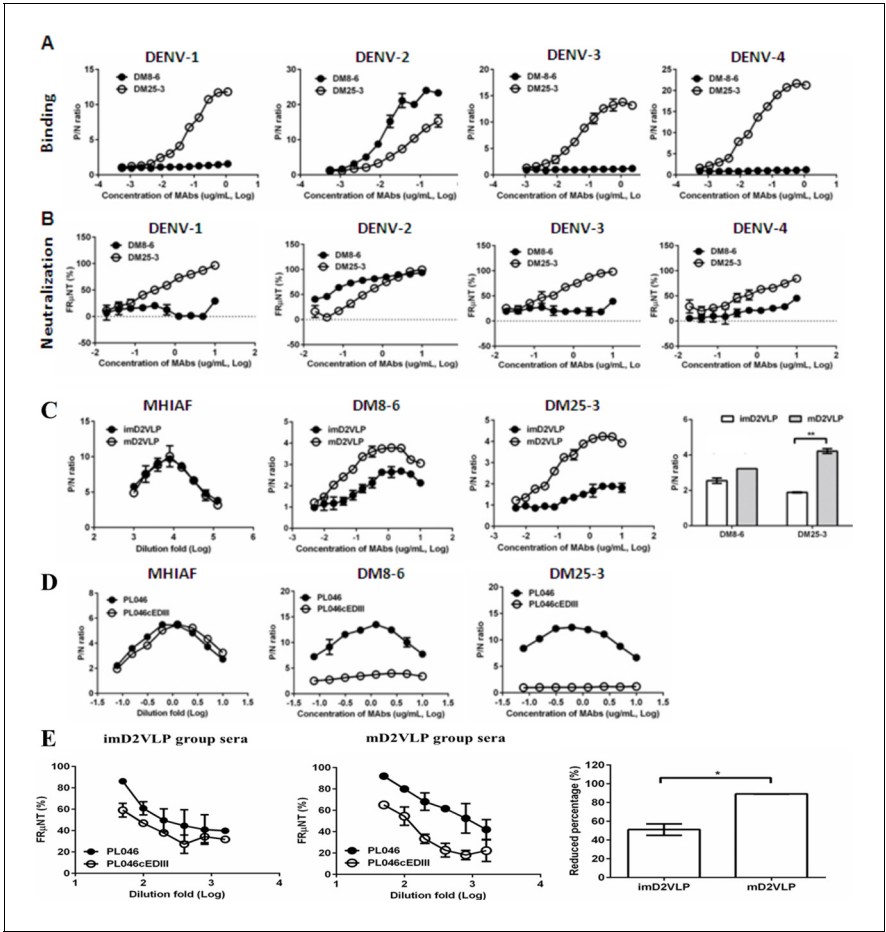

**Figure 6.** Characterization of murine monoclonal antibodies (MAbs) generated from mouse splenocytes following immunization with mD2VLP. Binding (**A**) and neutralizing (**B**) activities of MAbs DM8-6 and DM25-3 against DENV-1 to 4 were measured by ELISA and the focus-reduction micro-neutralization test (FRµNT). (**C**) Binding curves of MAb DM8-6 (center left) and DM25-3 (center right) against imD2VLP and mD2VLP were performed using ELISAs. Equal amount of both imD2VLP and mD2VLP was properly titrated by antigen-capture ELISA using mouse hyper-immune ascitic fluid (MHIAF) against DENV-2 (left). The difference in binding activities of both MAbs is presented by a bar graph (right). (**D**) A recombinant DENV-2 virus was produced by replacing domain III with a consensus sequence of domain III (PL046cEDIII) (*Fibriansah et al., 2015a*) and the binding activity of DM8-6 (center) and DM25-3 (right) was compared with that of parental DENV-2 strain PL046. Equal amount of both PL046 and PL046cEDIII was properly titrated by antigen-capture ELISA using mouse hyper-immune ascitic fluid (MHIAF) against DENV-2 (left). (**E**) FRµNT of two-fold diluted mice sera immunized with mD2VLP and imD2VLP against parental PL046 and PL046cEDIII DENV-2 viruses (n = 5 per group per experiment). The differences in FRµNT50 from mD2VLP and imD2VLP immunization groups were converted to bar chart at 1:1000 fold dilution of mice sera. The conversion was based on the formula 100*[FRµNT50 of (PL046- PL046cEDIII)/FRµNT50 of PL046]. P/N ratio refer to the antibody binding magnitude between designated VLP-containing (P) and VLP-free culture supernatant (N) by dividing the absorbance of P by that of N. The data are presented as means ± SEM from three independent experiments with two replicates. The two-tailed Mann-Whitney *U* test was used to test statistical significance. *p<0.05. **p<0.01.

DOI: https://doi.org/10.7554/eLife.38970.021

The following source data and figure supplements are available for figure 6:

**Source data 1.** source data for antibody mapping results in *Figure 6*.
DOI: https://doi.org/10.7554/eLife.38970.025

**Figure supplement 1.** Identification of the neutralizing epitopes of MAbs DM25-3 and DM8-6.
DOI: https://doi.org/10.7554/eLife.38970.022

**Figure supplement 2.** Transmission electron microscope (TEM) examination of negatively stained W101G-mutant mD2VLP.
DOI: https://doi.org/10.7554/eLife.38970.023

**Figure supplement 3.** Binding ELISA was performed to test the reactivity of sera from mice immunized with two doses of wtD2VLP, imD2VLP or mD2VLP against equal amounts of imD2VLP and fusion-peptide amino acid 101 mutant imD2VLP antigens (imD2VLP-W101G).
DOI: https://doi.org/10.7554/eLife.38970.024

understand the type of neutralizing antibodies induced by vaccination and establish the association between the level of such B-cell mediated immunological response and protection in the future study (*Katzelnick et al., 2017*; *Flipse and Smit, 2015*; *Henein et al., 2017*).

The major limitation of the current study is the unknown structure of imD2VLP or wtD2VLP to provide the proper explanation for the antibody response (*Figure 5*) or murine Mab mapping results (*Figure 4*). Our in silico footprint analysis showed the epitopes of Mab 1A1D-2 on DIII were well exposed on mD2VLP and was compared to DENV-2 virion (*Figure 3*), instead of imD2VLP. By comparing the structure of mD2VLP and D2 virion, it might be due to the groove within the E-protein dimer exposed several amino acids, including those located in both DI/II and DIII, previously suggested to be cryptic on virion (*Figure 3A*). However, our Mab mapping experiment was performed to compare between mD2VLP and imD2VLP. The results did confirm that several conformational-dependent Mabs, particularly those recognizing DI/II, were affected more by the structure of VLP with different maturity (*Figure 4*). The mapping results of Mab 1A1D-2 showing similar binding

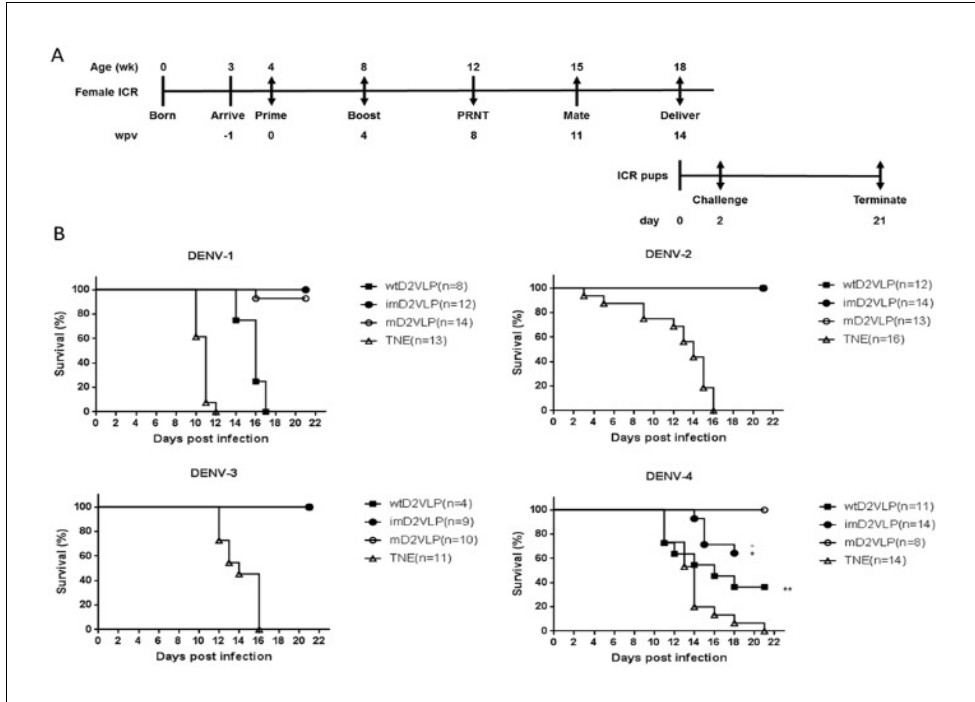

**Figure 7.** Schematic presentation of the schedule and survival curves for mouse immunization and challenge. (**A**) Groups of four 4-week-old, female, ICR mice were injected intramuscularly with imD2VLP and mD2VLP at week post vaccination (wpv) 0 and 4 at a dose of 4 μg/100 μL. Mice were bled from the retro-orbital sinus at week 4 following the second injection, and individual mouse serum collected from immunized females 1 week prior to mating was evaluated for the presence of the total IgG titer and the virus neutralization response by ELISA and focus-forming micro-neutralizing assay (FRNT). For the evaluation of passive protection by maternal antibody, ICR pups from the mating of non-immunized males with immunized females 11 weeks post initial vaccination were obtained for viral challenge. Pups from unvaccinated females were used as the challenge control. ICR pups from the designated groups were challenged individually through intracranial route at 2 days after birth with $10^4$ focus-forming unit (FFU) which were equivalent 141, 61, 11, 1000 times of 50% lethal doses (LD50) of DENV-1, to DENV-4, respectively. The percent survival of the mice was evaluated daily for up to 21 days. (**B**) Survival curve of pups delivered from the female mice receiving mD2VLP, imD2VLP monovalent vaccine or TNE control, then challenge with DENV-1 to 4 after birth. The in vivo protective efficacy of DENV-2 monovalent vaccine is maturity-dependent. N in parentheses indicated the numbers of pups of each group. Kaplan-Meier survival curves were analyzed by the log-rank test. * p<0.05, **p<0.01.

DOI: https://doi.org/10.7554/eLife.38970.026

The following source data is available for figure 7:

**Source data 1.** source data for antibody mapping results in *Figure 7*.
DOI: https://doi.org/10.7554/eLife.38970.027

activity to both mD2VLP and imD2VLP (*Figure 4D*) might indicate DIII was less affected by VLP maturity. This is consistent with the previous study in TBEV VLPs (*Lorenz et al., 2003*). Our current follow-up study is to solve the structure of imD2VLP, which might give a clear and direct picture of structural differences from Mab mapping.

Despite of size heterogeneity, our unique findings in this study showed that a mature, monovalent DENV-2 VLP with diameter of 31 nm possessing grooves within the E protein dimeric molecules on the surface of particles has the potential to induce highly protective, NtAbs against heterologous dengue viruses from all four serotypes. When combined delivery with other live attenuated vaccine, such as CYD-TDV or dengue tetravalent vaccine, there is potential to provide not just T-cell immunity, but also higher broad NtAb response through 'epitope-focusing' while exploring the next generation dengue vaccine in humans (*Crill et al., 2012*; *Rey et al., 2018*; *Flipse and Smit, 2015*). The strategy here may also provide a new direction for the development of other flavivirus vaccines, including Zika virus.

## Materials and methods

### Ethics statement

This study was carried out in compliance with the guidelines for the care and use of laboratory animals of the National Laboratory Animal Center, Taiwan. The animal use protocol has been reviewed and approved by the Institutional Animal Care and Use Committee (IACUC) of National Chung Hsing University (Approval Number: 101–58). All efforts were made to minimize suffering of mice.

### Viruses, cells and antibodies

The strains of DENV serotypes 1–4 used were Hawaii (Genbank accession: KM204119.1), 16681 (Genbank accession: KU725663.1), H87 (Genbank accession: KU050695.1) and H241 (Genbank accession: AY947539.1), respectively. COS-1 (ATCC: CRL 1650), Vero (ATCC: CRL 1587) and C6/36 cells (ATCC: CRL 1660) were grown in Dulbecco's modified Eagle's medium (DMEM, Gibco, Life Technologies, Grand Island, NY) and were supplemented with 10% heat-inactivated fetal bovine serum (FBS, Hyclone, ThermoFisher, MA), 0.1 mM nonessential amino acids (Gibco, Life Technologies, Grand Island, NY), 7.5% $NaHCO_3$, 100 U/ml penicillin, and 100 ug/ml streptomycin; 5% FBS was used for Vero cells. Cells were maintained at 37°C with 5% $CO_2$, except for C6/36 cells, which were maintained at 28°C without $CO_2$. All the cells used here are free of mycoplasma contamination checked regularly by following the commercial protocol (InvivoGen, Hong Kong). Parental DENV-2 strain PL046, generated from an infectious clone, and the domain III-swapped PL046 (PL046cEDIII), with domain III of DENV-2 replaced by consensus sequence as described previously (*Chen et al., 2013*; *Liang et al., 2009*), were provided by one of the co-authors, Dr. Y-L Lin.

Murine monoclonal antibody (MAb) 2H2, recognizing DENV prM protein, group cross-reactive antibodies (4G2, 4A1B-9, 6B3B-3, 6B6C-1, 5H3, 23–1, 23–2) recognizing E-protein of all four major pathogenic flavivirus serocomplexes; sub-group cross-reactive antibodies (T20, T5-1, T5-2, 1B7-5) recognizing more than one flavivirus serocomplexes; complex cross-reactive antibodies (D35C9, DB13-19) recognizing all four serotypes of DENV serocomplex viruses; sub-complex (1A1D-2, 9D12) recognizing more than one serotypes of DENV; serotype-specific MAb (3H5-1, 9A3D-8) recognizing DENV-2 only; rabbit sera and mouse hyper-immune ascitic fluid (MHIAF) for DENV-1 to 4 were provided by one of the authors, G-J Chang (DVBD, CDC, Fort Collins, CO). DENV-2 E protein-specific MAb DB32-6 DB25-2 and DB42-3 was provided by Dr. H-C Wu (Academia Sinica, TW), while DENV-2 prM-specific MAb 155–49 was obtained from H-Y Lei (National Cheng Kung University, Taiwan). MAb 2H2 was also labeled with biotin using EZ-Link Sulfo-NHS-Biotin kit (Thermo Fisher Scientific Inc., Rockford, IL) according to the manufacturer's instructions. In order to increase the recognition of M protein, anti-M antibody was produced by cloning the complete M protein sequence into pET21a (Novagen, Germany) and then purifying the expressed protein under denaturing conditions as described in the molecular cloning handbook. Ten micrograms (10 μg) of purified M protein with Freud's complete adjuvant was used to immunize the mice by the intraperitoneal route five times at 2 week intervals. Anti-mouse CD45R/B220 antibody conjugated with APC (allophycocyanin) and goat anti-mouse IgG conjugated with PE (phycoerythrin) were purchased from BioLegend (San Diego, CA).

Previously constructed and characterized recombinant plasmid pVD2, expressing the prM, 80% and 20% COOH terminus of the envelope proteins of DENV-2 (Asian one genotype, strain 16681) and Japanese encephalitis virus (strain SA14-14-2), respectively (*Galula et al., 2014*), was used in this study. The furin cleavage site of prM was mutated in pVD2 to generate a prM-uncleaved plasmid or as indicated in *Figure 1A* by using site-directed mutagenesis following the manufacturer's protocol (Stratagene, La Jolla, CA). The primers used for cloning and site-directed mutagenesis are provided in *Supplementary file 1*. Nucleotide sequencing confirmed that all the plasmids contained no other mutations other than those indicated.

## Sequence analysis of flavivirus prM sites

Flavivirus prM protein amino acid sequence alignments were performed using ClustalX 2.1 software with the representative strains and the GenBank accession numbers as follows: dengue virus serotype 2 (NP_056776), dengue virus serotype 1 (AIU47321), dengue virus serotype 3 (YP_001621843), dengue virus serotype 4 (NP_073286), Japanese encephalitis virus (NP_775664), St. Louis encephalitis virus (AIW82235), West Nile virus (AIO10814), tick-borne encephalitis virus (NP_775501), yellow fever virus (NP_041726), cell-fusion agent virus (NP_041725), Zika virus (BAP47441.1). Sequences of prM junction region were scored for their predicted cleavability by furin using the PiTou 2.0 software package (*Tian et al., 2012*). A negative score indicates a sequence predicted not to be cleaved by furin, whereas a positive score denotes prediction of furin cleavability.

## Antigen production and purification

To produce virus-like particle (VLP) antigens, COS-1 cells at a density of $1.5 \times 10^7$ cells/mL were electroporated with 30 µg of each pVD2 plasmid following the previously described protocol (*Chang et al., 2000*). After electroporation, cells were seeded into 75 cm$^2$ culture flasks (Corning Inc., Corning, NY, USA) containing 15 mL growth medium and allowed to recover overnight at 37℃. The growth medium was replaced the next day with a maintenance medium containing serum-free medium (SFM4MegaVirTM, SH30587.01, Hyclone, ThermoFisher) supplemented with non-essential amino acid (NEAA), GlutaMAX, sodium pyruvate and cholesterol (Gibco, Life Technologies, Grand Island, NY), and cells were continuously incubated at 28℃ with 5% $CO_2$ for VLP secretion. Tissue-culture media were harvested 3 days after transfection and clarified by centrifugation at $8,000 \times g$ for 30 min (mins) at 4℃ in AF-5004CA rotor (Kubota, Tokyo, Japan) using a Kubota 3740 centrifuge. The harvested media were first concentrated 20-fold using 100K Amicon Ultra centrifugal filters (Merck Millpore, CA) before loading onto a 20% sucrose cushion and concentrated by ultracentrifugation at 28,000 rpm for 16 hr at 4℃ in a Beckman SW28 rotor. Purified VLPs were resuspended in 250 µl TNE buffer (50 mM Tris-HCl, 100 mM NaCl, 0.1 mM EDTA, pH 7.4) for every liter of harvested medium at 4℃ overnight. The VLPs were further purified by rate zonal centrifugation in a 5% to 25% sucrose gradient at 25,000 rpm at 4℃ for 3 hr. All gradients were made with TNE buffer and were centrifuged in a Beckman SW41 rotor. Fractions of 0.5 ml were collected by upward displacement and assayed by antigen-capture ELISA. For experiments that required highly purified VLPs, proteins with the peak OD values from antigen-capture ELISA were pelleted at 40,000 rpm at 4℃ for 4 hr using a Beckman SW41 rotor, and re-suspended in 250 µl TNE buffer. The protein concentration was measured by the Bradford assay (BioRad, Hercules, CA) following the commercial protocol and using bovine serum albumin (BSA, New England Biolabs, MA) as a standard. Purified VLPs were also labeled with fluorescein for antigen-specific B-cell sorting following the manufacture's protocol (*Zhang et al., 2010*).

## ELISA

The ratio of prM to E protein was determined using DENV-2 immune rabbit serum to capture VLPs. The DENV-2 E and prM proteins were measured by ELISA using MAb 3H5 (specific for DENV-2 domain III) and MAb 155–49 (specific for DENV prM). The ratio was calculated as absorbance for prM/absorbance for E protein. The uncleaved prM VLP (imD2VLP), which contained amino acid mutations at P1 and P2 sites from amino acid residue R/K to T/S (*Li et al., 2008*), respectively, was used as a standard to calculate the percentage of prM cleavage. Percent cleavage of prM was then calculated with reference to imD2VLP, which was assumed to be 100% uncleaved, as previously described (*Dejnirattisai et al., 2015*).

Antigen-capture ELISA was performed to quantify the amount of different VLP antigens. Briefly, flat-bottom 96-well MaxiSorp NUNC-Immuno plates (NUNC, Roskilde, Denmark) were coated with 50 μL of rabbit anti-DENV-2 VLP serum at 1:500 in bicarbonate buffer (0.015 M $Na_2CO_3$, 0.035 $NaHCO_3$, pH 9.6), incubated overnight at 4°C, and blocked with 200 μL of 1% BSA in PBS (1% PBSB) for 1 hr at 37°C. Clarified antigens were titrated two-fold in PBSB, and 50 μL of each dilution was added to wells in duplicate, incubated for 2 hr at 37°C, and washed five times with 200 μL of 1x PBS with 0.1% Tween-20 (0.1% PBST). Normal COS-1 cell tissue culture fluid and cell pellets were used as control antigens. Captured antigens were detected by adding 50 μL of anti-DENV-2 MHIAF at 1:2000 in blocking buffer, incubated for 1 hr at 37°C, and washed for five times. Fifty microliters of HRP-conjugated goat anti-mouse IgG (Jackson ImmunoResearch, Westgrove, PA, USA) at 1:5000 in blocking buffer was added to wells and incubated for 1 hr at 37°C to detect MHIAF. Subsequently, plates were washed ten times. Bound conjugate was detected with 3,3',5,5'-tetramethylbenzidine substrate (Enhanced K-Blue TMB, NEOGEN Corp., Lexington, KY, USA), after incubation at room temperature for 10 mins, and addition of 2N $H_2SO_4$ to stop the reaction. Reactions were measured at $A_{450}$ using a Sunrise TECAN microplate reader (Tecan, Grödig, Austria). The capability of antigen-capture ELISA to detect imD2VLP or mD2VLP was measured against purified D2VLP at known total protein concentration from individual preparations. Data are expressed as P/N ratio by dividing the OD450 value from each dilution of mouse sera or MAb by the OD450 value from the control COS-1 culture supernatant.

An IgG ELISA was used to assay the presence of antigen-specific IgG in the post-vaccination mouse sera using the same antigen-capture ELISA protocol described above with minor modifications. Equal amounts of purified antigens were added into each well of an Ag-capture ELISA plate. The concentration of purified antigens were determined from the standard curves generated using a sigmoidal dose-response analysis using GraphPad Prism (version 6.0, GraphPad Software, Inc., La Jolla, CA, USA). Individual mouse sera, collected from mice which had the same immunization schedule, initially diluted at 1:1,000, were titrated two-fold and added into wells in duplicate, and were incubated for 1 hr at 37°C. Pre-vaccination mouse sera were used as negative controls. Incubations with conjugate and substrate were carried out according to the standard Ag-capture ELISA as outlined. The $OD_{450}$ values, modeled as non-linear functions of the log10 serum dilutions using a sigmoidal dose-response (variable slope) equation and endpoint antibody titers from two independent experiments, were determined as the dilutions where the OD value was twice the average OD of a negative control.

Epitope-blocking ELISAs were performed to determine the vaccinated mouse response to the prM protein. The setup was similar to the IgG ELISA wherein plates were coated with rabbit anti-DENV-2 VLP serum, and blocked with 1% BSA in PBS. After washing, pooled mouse sera were diluted two-fold in blocking buffer starting from 1:1,000, and were then incubated with imD2VLP antigen (pre-titrated to $OD_{450}$ = 1.0) for 1 hr at 37°C. After serum incubation and washing, MAb 2H2 conjugated with biotin by EZ-Link Sulfo-NHS-Biotin (ThermoFisher, CA) at 1:4000 dilution was added to each well and incubated for 1 hr at 37°C to compete with the already-bound antibody from the immune mouse sera specific for the imD2VLP antigen. Bound MAb 2H2 conjugate was detected with 1:1000 HRP-conjugated streptavidin (016-030-084, Jackson ImmunoResearch), and was incubated for 1 hr at 37°C. After washing with PBS for ten times, TMB substrate was added into the wells and the plates were incubated for 10 mins, and the reaction was stopped with 2 N $H_2SO_4$. Reactions were measured at $A_{450}$. Percent blocking was determined by comparing replicate wells with Biotin-conjugated MAb competing against pre-adsorbed naïve mouse serum using the formula: % Blocking = [OD450 of imD2VLP-$OD_{450}$ of imD2VLP blocked by MAb 2H2)/$OD_{450}$ of imD2VLP] × 100.

Binding-ELISAs were used to assess the binding activity of MAbs or mouse immune sera to D2VLP or mutant antigens using a similar antigen-capture ELISA set-up, except that two-fold dilutions of the specific MAb or immune mouse sera replaced the anti-DENV-2 MHIAF. Equal amounts of D2VLP antigens were added into wells, and were standardized using purified D2VLPs. The antibody endpoint reactivity was determined in a similar manner to the determination of antigen endpoint secretion titers.

## SDS-PAGE and western blotting

Equal amounts of purified VLPs and sample buffer were mixed and analyzed by 12% Tricine-sodium dodecyl sulfate-polyacrylamide gel electrophoresis (Tricine-SDS-PAGE) (*Schägger, 2006*). For

immunodetection, proteins were blotted from gels onto nitrocellulose membranes (iBlot2NC mini stacks, ThermoFisher Scientific) with iBlot 2 Gel Transfer Device (ThermoFisher Scientific). The membranes were incubated for 1 hr at room temperature in phosphate-buffered saline (pH 7.4) containing 5% skim milk (BD biosciences, CA) to block nonspecific antibody binding. After 1 hr incubation, membranes were individually stained to detect E, prM and M proteins by using anti-DENV2 MHIAF at 1:2000, anti-DENV prM MAb 155–49 at 0.5 µg/ml and mouse anti-M sera at 1:25, respectively, at 4°C overnight. Membranes were washed three times for 15 min each. DENV-specific bound immunoglobulin was recognized with HRP-labeled goat anti-mouse IgG (Jackson ImmunoReasearch, PA), and was visualized with ECL (enhanced chemiluminescent substrate, GE Healthcare, UK) according to the manufacturer's protocol. The ECL signals were detected by ImageQuant$^{TM}$ LAS 4000 mini (GE Healthcare). However, M protein was visualized using TMB membrane peroxidase substrate (KPL, MD) to avoid high background from ECL.

## Mouse experiment

Groups of four 4-week-old female BALB/c mice were injected intramuscularly with imD2VLP and mD2VLP at weeks 0 and 4 at a dose of 4 µg/100 µL PBS divided between the right and left quadriceps muscle. Mice were bled from the retro-orbital sinus at week 4 following the second injection, and individual serum specimens were evaluated for DENV-2 specific antibodies by ELISA and focus-reduction micro-neutralization test (FRµNT), as described in the following section.

For the evaluation of passive protection by maternal antibody, ICR pups from the mating of non-immunized males with immunized females 11 weeks post initial vaccination were obtained for viral challenge. Immune sera were collected from immunized females 1 week prior to mating to confirm the presence of the total IgG titer as well as the virus neutralization antibody titer. Pups from unvaccinated females were used as the challenge control. ICR pups from the designated groups were challenged individually through the intracranial route at 2 days after birth with $10^4$ focus-forming units (FFU) which were equivalent to 141, 61, 11, 1000-fold of 50% lethal doses (LD50) of DENV-1 (strain Hawaii), DENV-2 (strain 16681), DENV-3 (strain H87) and DENV-4 (strain BC71/94, kindly provided by one of the co-author, Dr. Chang from US-CDC), respectively. Mouse survival was evaluated daily for up to 21 days.

## Virus neutralization

The neutralizing ability of the immune mouse sera for all serotypes of DENV was measured by focus-reduction micro neutralization test (FRµNT), as previously described (*Galula et al., 2014*). Briefly, $2.475 \times 10^4$ Vero cells/well were seeded onto flat-bottom 96-well Costar cell culture plates (Corning Inc., Corning, NY, USA) and incubated for 16 hr overnight at 37°C with 5% CO$_2$. Pooled sera were initially diluted at 1:10, heat-inactivated for 30 min at 56°C, titrated two-fold to a 40 µL volume, and 320 pfu/40 µL of DENV-1 to 4 was added to each dilution. The mixtures were then incubated for 1 hr at 37°C. After incubation, 25 µL of the immune complexes were added in duplicates onto plates containing Vero cell monolayers. Plates were incubated for 1 hr at 37°C with 5% CO$_2$ and rocked every 10 mins to allow infection. Overlay medium containing 1% methylcellulose (Sigma-Aldrich Inc., St. Louis, MO, USA) in DMEM with 2% FBS was added, and plates were incubated at 37°C with 5% CO$_2$. Forty-eight hours later, plates were washed, fixed with 75% acetone in PBS and air-dried. Immunostaining was performed by adding serotype-specific MHIAF at 1:600 in 5% milk and 0.1% PBST and incubated for 60 min at 37°C. Plates were washed and goat anti-mouse IgG-HRP at 1:100 in 5% milk and 0.1% PBST was added; plates were incubated for 45 min at 37°C. Infection foci were visualized using a peroxidase substrate kit, VectorVIP SK-4600 (Vector Laboratories, Inc., Burlingame, CA, USA), following the manufacturer's instructions. FRµNT titers were calculated for each virus relative to a virus only control back-titration. Titers were determined as a 50% reduction of infection foci (FRµNT50) and were modeled using a sigmoidal dose-response (variable slope) formula. All values were determined from the average of two independent experiments. Target virus strains were: DENV-1, Hawaii; DENV-2, 16681; DENV-3, H87; and DENV-4, H241. For E-dimer, inter-domain-specific neutralizing antibodies, the recombinant parental DENV-2 strain PL046 and EDIII-swapped PL046cEDIII were used as target viruses. In the calculation of geometric mean titers (GMT) for graphic display and statistical analysis, a FRµNT50 titer of <10 was represented with the value of 1 and 5, respectively.

## Generating hybridomas and MAb screening

Hybridomas secreting anti-DENV antibodies were generated from the mD2VLP immunized mice according to a standard procedure (*Köhler and Milstein, 1975*), with slight modifications (*Chen et al., 2007*). First, the mD2VLP-immunized mouse was boosted with another 4 µg of mD2VLP 24 weeks prior to terminal bleeding. At day 4 after the third immunization, splenocytes were harvested from the immunized mouse and fused with NSI/1-Ag4-1 myeloma cells using an antibody delivery kit following manufacturer's recommendations (GenomONETM-CF HVJ Envelope Cell Fusion Kit, Gosmo Bio Co, ISK10 MA17). Fused cell pellets were suspended in DMEM supplemented with 15% FBS, hypoxanthine-aminopterin-thymidine medium, and hybridoma cloning factor (ICN, Aurora, OH). Hybridoma colonies were screened for secretion of MAbs by ELISA following the procedures as described above. However, the cell culture supernatant of C6/36 cells infected with DENV-2 virus at a multiplicity of infection (moi) equal to 1.0 was used as the antigens and the culture supernatant from each of the hybridoma colonies was used as the detecting antibody. Selected positive clones were subcloned by limiting dilution. Ascitic fluids were produced in pristane-primed BALB/c mice. Hybridoma cell lines were grown in DMEM with 10% heat inactivated FBS. MAbs were affinity purified with protein G Sepharose 4B gel, and the amount of each purified MAb was quantified by comparison with a known amount of IgG used in a standard ELISA.

## Immunogold labeling and transmission electron microscopy (TEM)

To detect mature DENV-2 VLPs (mD2VLPs) as spherical particles, 3 µL of freshly prepared samples were adsorbed onto a glow-discharged nickel grid (EMS CF-200-Ni) for 20 mins, The grid containing the samples was incubated in 1% BSA/PBS blocking buffer for 1 hr at room temperature. As soon as the excess liquid was removed with a filter paper, DENV-2-specific MAb DB32-6 was used as the primary antibody (dilution 1:500), and was added to the samples for 2 hr at room temperature. Then the grid was washed with blocking buffer and incubated with 6 nm nanogold-conjugated Donkey anti-mouse IgG (Abcam, MA) (dilution 1:30) at room temperature for 1 hr. The excess liquid was removed with filter paper and fixed with 1% glutaraldehyde in PBS buffer (GA/PBS) for 10 mins. After fixing, excess liquid was removed using a filter paper. Finally, the sample was stained with 2% uranyl acetate (UA) for 1 min and air-dried. The immunogold labeling process was performed in a high humidity chamber. The immunogold labeled mD2VLPs were inspected by TEM. The images were taken by a JEM1400 electron transmission microscope at a magnification of 100,000x using a 4k × 4 k Gatan 895 CCD camera. The diameters of particles were measured by ImageJ software.

## Cryo-EM and 3d reconstruction

A freshly prepared dengue virus sample (3 µl) was placed onto a glow-discharged Quatifoil 2/2 grid (Quatifoil GmbH, Germany), blotted with filter paper, and plunged into liquid nitrogen-cooled liquid ethane using Gatan CP3. Cryo-EM images were recorded with a JEM2100F using an accelerating voltage of 200kV and a magnification of 15,000x using a direct electron detector (DE-12 Camera System - Direct Electron, LP) with a 6 µm pixel size (corresponding to ~4 Å at the specimen level). The measured defocus values of these images range from −2 µm to −4.5 µm. The imaging electron dosage was ~10 e⁻/Å (*Bhatt et al., 2013*).

Variations in size and shape were observed in the EM images (*Figure 2—figure supplement 1A, B*). The irregularly-shaped particles were eliminated through visual inspection, while the spherical particles were subjected to further image analyses. The particle size analyses showed that the major peaks were located at ~26 nm,~31 nm and ~36 nm diameter size classes which were therefore denoted by small ('S'), medium ('M') and large ('L'). The heterogeneity analyses by EMAN2 (*Tang et al., 2007*) and XMIPP (*Sorzano et al., 2004*; *de la Rosa-Trevín et al., 2013*) showed that all of the classes had distinct two layers and that the particles classes with diameters of 31 nm had more prominent features than others (*Figure 2—figure supplement 1B,C*). Taken together with the immunogold labeling results, we selected the particles with size of ~31 nm or further 3D reconstruction process. The structure reconstruction processes were performed by EMAN2. There were 4217 spherical particles with the size of ~31 nm which were included in the reconstruction process and the icosahedral symmetry was imposed during the whole process. The reconstruction process ended when there was no improvement achieved. The resolution of the final reconstructions was 13.1 Å from a Fourier shell correlation curve using the gold standard resolution estimate in EMAN2

(*Figure 2—figure supplement 3*). Solvent accessible surface area (SASA) of individual amino acid molecules on mD2VLP was calculated by POPS program (*Fornili et al., 2012*; *Cavallo et al., 2003*).

## Fitting

The atomic structure of mD2VLP was modeled based on the cryo-EM structure of the mature dengue virus at 3.5 Å resolution (PDB ID: 3J27) using MODELLER 9v15 (*Eswar et al., 2007*). The envelop proteins were fitted rigidly using the Fit-in-Map tool in *UCSF Chimera*, when the CCC (Cross-Correlation Coefficient) score was maximized, the complete copies were generated to follow the T = 1 arrangement with *Multiscale Models* using Icosahedral symmetry, xyz 2-fold axes (VIPER).

## Statistical analysis

All data are represented as means ± standard error, and analyzed using GraphPad Prism version 6.0. Unpaired t test was used to analyze data sets between two groups. Mann-Whitney U test to account for non-normality of some transformed data was also applied. P values < 0.05 were considered significant.

## Data and materials availability

The cryo-EM density map of mD2VLP has been deposited to Electron Microscopy Data Bank under accession number EMDB6926.

## Acknowledgments

We thank Dr. Felix Rey for commenting on the structure of dengue VLP and Ann Hunt for English editing. This study was supported by Ministry of Science and Technology Taiwan (MOST 104-2320-B-006-027, MOST 105-2320-B-006-017-MY3, MOST-104-2633-B-005-001 and MOST 106-2313-B-005-029)

## Additional information

### Funding

| Funder | Grant reference number | Author |
| --- | --- | --- |
| Ministry of Science and Technology, Taiwan | 104-2320-B-006-027 | Shang-Rung Wu |
| Ministry of Science and Technology, Taiwan | 105-2320-B-006-017-MY3 | Shang-Rung Wu |
| Ministry of Science and Technology, Taiwan | 104-2633-B-005-001 | Day-Yu Chao |
| Ministry of Science and Technology, Taiwan | MOST 106-2313-B-005-029 | Jyung-Hurng Liu |

The funders had no role in study design, data collection and interpretation, or the decision to submit the work for publication.

### Author contributions

Wen-Fan Shen, Jedhan Ucat Galula, Data curation, Investigation, Visualization; Jyung-Hurng Liu, Software, Formal analysis, Visualization; Mei-Ying Liao, Sheng-Ren Chen, Data curation, Formal analysis, Investigation; Cheng-Hao Huang, Yu-Chun Wang, Matthew T Whitney, Data curation, Investigation; Han-Chung Wu, Resources, Methodology; Jian-Jong Liang, Resources, Investigation; Yi-Ling Lin, Resources; Gwong-Jen J Chang, Resources, Supervision, Writing—review and editing; Shang-Rung Wu, Conceptualization, Data curation, Software, Supervision, Funding acquisition, Validation, Investigation, Visualization, Writing—original draft, Project administration, Writing—review and editing; Day-Yu Chao, Conceptualization, Resources, Data curation, Supervision, Funding acquisition, Validation, Investigation, Methodology, Writing—original draft, Project administration, Writing—review and editing

## Author ORCIDs

Jyung-Hurng Liu (iD) http://orcid.org/0000-0002-7173-0372
Han-Chung Wu (iD) http://orcid.org/0000-0002-5185-1169
Gwong-Jen J Chang (iD) http://orcid.org/0000-0001-9959-6585
Shang-Rung Wu (iD) http://orcid.org/0000-0003-1940-7612
Day-Yu Chao (iD) http://orcid.org/0000-0001-7139-026X

## Ethics

Animal experimentation: This study was carried out in compliance with the guidelines for the care and use of laboratory animals of the National Laboratory Animal Center, Taiwan. The animal use protocol has been reviewed and approved by the Institutional Animal Care and Use Committee (IACUC) of National Chung Hsing University (Approval Number: 101-58). All efforts were made to minimize suffering of mice.

## Decision letter and Author response

Decision letter https://doi.org/10.7554/eLife.38970.033
Author response https://doi.org/10.7554/eLife.38970.034

# Additional files

## Supplementary files

• Supplementary file 1. Nucleotide sequences of primers for site-directed mutagenesis used in this study
DOI: https://doi.org/10.7554/eLife.38970.028

• Transparent reporting form
DOI: https://doi.org/10.7554/eLife.38970.029

## Data availability

The cryo-EM density map of mD2VLP has been deposited to Electron Microscopy Data Bank under accession number EMD-6926. All data generated or analyzed during this study are included in the manuscript and supporting files.

The following dataset was generated:

| Author(s) | Year | Dataset title | Dataset URL | Database and Identifier |
| --- | --- | --- | --- | --- |
| Shang-Rung Wu, Day-Yu Chao | 2018 | CryoEM structure of mature dengue virus-like particle | http://emsearch.rutgers.edu/atlas/6926_summary.html | Electron Microscopy Data Bank, EMD-6926 |

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
