## [Decision Letter]

Thank you for submitting your article "An epitope-resurfaced virus-like particle can induce broad neutralizing antibody against four serotypes of dengue virus" for consideration by *eLife*. Your article has been reviewed by Arup Chakraborty as the Senior Editor and Reviewing Editor and two reviewers. The reviewers have opted to remain anonymous.

The reviewers have discussed the reviews with one another and the Reviewing Editor has drafted this decision to help you prepare a revised submission.

Summary:

A central problem with current approaches to vaccination against dengue virus (DENV) is that current vaccine formulations elicit antibody responses to both prM and M forms of the viral envelope (E) protein. Anti-prM antibodies enhance disease pathogenesis by DENV and therefore represent a serious concern for DENV vaccine concepts. To address this problem, the investigators of this study have presented E protein on an immunogenic VLP-array and at the same time have manipulated the furin cleavage site of this envelope protein to substantially increase cleavage and ensure the presentation of the M form only (=mD2VLP). In contrast to VLP forms where E was not so manipulated, mD2VLP was more effective in mouse-DENV challenge models and could stimulate antibody responses with broader neutralizing properties. The authors engineered VLPs with different levels of prM and analyzed their epitopes recognized by known monoclonal antibodies and the antibody response elicited these VLPs with regard to epitope recognition and in vivo protection. The authors attempt to relate the structure of the mature VLP to its superior ability especially the "open" configuration of the E dimer in antibody accessibility and the ability to elicit better neutralization. The furin engineering concept represents an important advance in the field for advancing next generation DENV vaccines, and the study is largely well executed. While the authors' overall conclusions are likely correct, the VLP vaccine preparations were extremely heterogeneous, meaning that much of the structure-based conclusions on the vaccine preparation have been overstated and are quite misleading. These and other concerns that need to be addressed are noted below.

Essential revisions:

1) A central concern is that the heterogeneity of the mD2VLP prep does not seem to be properly acknowledged. The title for example "an epitope-resurfaced virus-like particle…" or in the abstract "mD2VLP particles possess a T=1 icosahedral symmetry with a groove located within the E-protein dimers near the 2-fold vertices that exposed highly overlapping, cryptic neutralizing epitopes through cryo-electron microscopy reconstruction" are not reflective of the actual vaccine composition. There are many examples of uniform VLP-vaccine preps, but this does not appear to be one of them. Notably, there is considerable variation in the diameter and morphology of the VLPs (Figure 1—figure supplement 1, Figure 1—figure supplement 2, Figure 2—figure supplement 2). Then a follow up cryo em structure appears to have been selectively picked, but it does that does not seem to reflect the bulk population. Also, the structure of the other two forms of VLPs (wtD2VLP and imD2VLP) was not experimentally determined. In the literature, there are examples of particle vaccines that do pass uniformity tests, however the one reported in the paper does not. It is misleading to then claim to have a structure that then supplies certain downstream immunological effects. This is not to say non-uniform vaccines aren't worth having or making, especially those that provide efficacy. But, the structural interpretations and predictions (Figure 2, Figure 3, Figure 2—figure supplement 3, Figure 3—figure supplement 1, Figure 3—figure supplement 2) need to be toned down or removed because they are not representative of the vaccine preparation as a whole.

2) It is not immediately clear why Figure 7 is important. Why is it important to know which murine antibody families are used? If important, the DENV B cell probe needs to be properly validated; i.e. a pre-immune flow plot should be shown to demonstrate that the antibodies pulled out actually bind DENV. The DENV positive gate seems to include B cells that are likely not antigen specific.

3a) A number of findings that seem difficult to explain were not discussed in the manuscript. For example, in Figure 4, binding of 1A1-D2 (which recognizes a cryptic epitope of DIII which is supposedly more exposed in mD2VLP) to imD2VLP and mD2VLP was similar. This is not consistent with the main argument.

b) If one follows the authors' assertion that the antibody response was determined by the VLPs' structure and epitope accessibility, immunization with imD2VLPs should elicit primarily antibody responses to the conformational independent epitope of the fusion peptide and perhaps a minor response to other epitopes on DIII and should be poorly neutralizing. However, the neutralizing activity of the imD2VLP immune sera was comparable to wtD2VLP immune sera to DEN-2, DEN-3, and DEN-4, and not different from mD2VLP immune sera to DEN-3 (Figure 5C). The difference in neutralizing activity elicited by these VLPs to various serotypes of DENV both in vitro and in vivo was also not explained and discussed.

[Editors' note: further revisions were requested prior to acceptance, as described below.]

Thank you for resubmitting your work entitled "An epitope-resurfaced virus-like particle potentially induce broad neutralizing antibodies against dengue viruses" for further consideration at *eLife*. Your revised article has been favorably evaluated by Arup Chakraborty (Senior Editor and handling Reviewing Editor), and two reviewers.

Unfortunately, we do not think you have made significant changes in response to the major criticism regarding the size heterogeneity that does not support the structure related arguments made in the manuscript. The field works extremely hard to generate uniform structurally refocused immunogens and it is a real achievement when this occurs. This has not been achieved here. Claiming in the title "An epitope-resurfaced virus-like particle…" means something with respect to structure-based vaccine design – namely that a single re-engineered entity has been developed. But, you have have generated a preparation with reconfigured epitopes that is extremely heterogenous in size and morphology. However, given the problem of disease enhancement by anti-prM antibodies, the furin engineering concept represents an important advance worthy of publication in this journal.

Essential revisions:

1) The title needs revising to e.g. "Epitope resurfacing on VLP vaccine preparation to…"

2) A clear acknowledgement of the heterogeneity from the outset, in the abstract and beyond. E.g. in abstract "herein we show for the first time that mD2VLP particles possess a T=1 icosahedral symmetry with a groove located within the E 52protein dimers near the 2-fold vertices that exposed highly overlapping, cryptic neutralizing 53 epitopes through cryo-electron microscopy reconstruction". This does not seem to be accurate. The mD2VLP preparation consist of entities that are mostly not this. The authors need to state specifically that the structure presented can guide vaccine design, but likely does not represent what is precisely in the vaccine.

---

## [Author Response]

Essential revisions:1) A central concern is that the heterogeneity of the mD2VLP prep does not seem to be properly acknowledged. The title for example "an epitope-resurfaced virus-like particle…" or in the abstract "mD2VLP particles possess a T=1 icosahedral symmetry with a groove located within the E-protein dimers near the 2-fold vertices that exposed highly overlapping, cryptic neutralizing epitopes through cryo-electron microscopy reconstruction" are not reflective of the actual vaccine composition. There are many examples of uniform VLP-vaccine preps, but this does not appear to be one of them. Notably, there is considerable variation in the diameter and morphology of the VLPs (Figure 1—figure supplement 1, Figure 1—figure supplement 2, Figure 2—figure supplement 2). Then a follow up cryo em structure appears to have been selectively picked, but it does that does not seem to reflect the bulk population. Also, the structure of the other two forms of VLPs (wtD2VLP and imD2VLP) was not experimentally determined. In the literature, there are examples of particle vaccines that do pass uniformity tests, however the one reported in the paper does not. It is misleading to then claim to have a structure that then supplies certain downstream immunological effects. This is not to say non-uniform vaccines aren't worth having or making, especially those that provide efficacy. But, the structural interpretations and predictions (Figure 2, Figure 3, Figure 2—figure supplement 3, Figure 3—figure supplement 1, Figure 3—figure supplement 2) need to be toned down or removed because they are not representative of the vaccine preparation as a whole.

The size heterogeneity of VLP is commonly found in flavivirus including tick-borne encephalitis virus (TBEV) or West Nile virus (WNV). In TBEV study, VLP preparation showed a range of size distribution with the majority fell into 31nm as mature particles. However, WNV VLPs were secreted as large (40-50nm) and small (20-30nm) particle sizes, which show different degrees of maturity and immunogenicity. We agree with the reviewers and editors’ general comments that ex vivoprotection elicited by VLPs immunization cannot solely attribute to the protective immunity induced by 30nm-VLP due to the size heterogeneity of our preparation. We have toned down the conclusion and have revised in the manuscript including the Title and Discussion section.

2) It is not immediately clear why Figure 7 is important. Why is it important to know which murine antibody families are used? If important, the DENV B cell probe needs to be properly validated; i.e. a pre-immune flow plot should be shown to demonstrate that the antibodies pulled out actually bind DENV. The DENV positive gate seems to include B cells that are likely not antigen specific.

The initial intention of the experiment performed on Figure 7 is to find out if the CR Nt Mab identified from hybridoma fused between myeloid cell and splenocytes of mice immunized by mature VLP is due to clonal expansion. Although we cannot completely exclude the possibility of sorting non-DENV-specific B cell into the wells, the gating of either B220-positive or D2VLP-positive was done separately using control mouse splenocytes. We are currently undertaking the follow-up study to express heavy chain and light chain of DENV B cell clones. Recombinant antibodies will be used to compare their binding properties with the MAbs generated from hybridomas. Furthermore, we agree this portion of our manuscript cannot be substantiated and elect to remove the portion with Figure 7 from the manuscript.

3a) A number of findings that seem difficult to explain were not discussed in the manuscript. For example, in Figure 4, binding of 1A1-D2 (which recognizes a cryptic epitope of DIII which is supposedly more exposed in mD2VLP) to imD2VLP and mD2VLP was similar. This is not consistent with the main argument.

First, our best guess is that DIII is less affected by maturity of VLP and that’s the reason why the 1A1D-2 mapping results showed similar binding to three different D2VLPs. Second, based on the prediction of epitope accessibility between VLP and virion, the residues surrounding 1A1D-2 binding have overall higher surface/solvent accessibility in mD2VLPs than DENV-2 virions. We, thus, hypothesize that the use of mD2VLPs as immunogen are more accessible than imD2VLPs or wtD2 VLPs to naïve B-cells. However, the epitope-mapping used in this study may not be sufficiently sensitive to distinguish the difference in surface accessibility between mD2VLPs and imD2VLPs. Also, we don’t have the structure of imD2VLP. The prediction using DENV-2 immature virion, which was solved by Yu et al., cannot be directly extrapolate to imD2VLP. We are attempting to solve the structure of imD2VLP, and hopefully we can address the question directly. Overall our finding from Figure 4 is that the binding of several conformational-dependent MAbs are affected by the structural differences between mD2VLP and imD2VLP, as suggested from TBEV VLP studies. We have revised accordingly in Discussion section.

b) If one follows the authors' assertion that the antibody response was determined by the VLPs' structure and epitope accessibility, immunization with imD2VLPs should elicit primarily antibody responses to the conformational independent epitope of the fusion peptide and perhaps a minor response to other epitopes on DIII and should be poorly neutralizing. However, the neutralizing activity of the imD2VLP immune sera was comparable to wtD2VLP immune sera to DEN-2, DEN-3, and DEN-4, and not different from mD2VLP immune sera to DEN-3 (Figure 5C). The difference in neutralizing activity elicited by these VLPs to various serotypes of DENV both in vitro and in vivo was also not explained and discussed.

In our study, we showed that imD2VLP did induce significantly lower proportion of conformation-dependent epitope fusion peptide amino acid 101-specific antibody (Figure 6C) and antibodies recognizing DIII-conformational-independent epitopes (Figure 6E) compared to mD2VLP. Thus, theoretically, the immune response elicited by imD2VLP should be poorly neutralizing, as pointed out by the reviewers. However, the discrepancy of the results from neutralizing antibodies and the proportion of epitope-specific antibodies elicited by imD2VLP and D2VLP could be explained that other CR epitopes-recognizing conformation-independent antibodies such as 4G2-like antibodies, though poorly neutralizing, can still neutralize DENV while in a large quantity. To confirm this argument, we mutated amino acid 101 on fusion peptide of imD2VLP and compared the loss of binding of antibodies from the immuned mice sera with the wild-type imD2VLP (Figure 5—figure supplement 4). Mutation of amino acid 101 on imD2VLP disrupted the binding of murine Mab 4G2. Meanwhile, the results showed that the proportion of 4G2-like antibodies among the sera receiving imD2VLP immunization was greater than that of sera either receiving mD2VLP or wtD2VLP. We agree with the reviewers and have added one paragraph in Discussion section in the revised manuscript.

[Editors' note: further revisions were requested prior to acceptance, as described below.]

Unfortunately, we do not think you have made significant changes in response to the major criticism regarding the size heterogeneity that does not support the structure related arguments made in the manuscript. The field works extremely hard to generate uniform structurally refocused immunogens and it is a real achievement when this occurs. This has not been achieved here. Claiming in the title "An epitope-resurfaced virus-like particle…" means something with respect to structure-based vaccine design – namely that a single re-engineered entity has been developed. But, you have have generated a preparation with reconfigured epitopes that is extremely heterogenous in size and morphology. However, given the problem of disease enhancement by anti-prM antibodies, the furin engineering concept represents an important advance worthy of publication in this journal.Essential revisions:1) The title needs revising to e.g. "Epitope resurfacing on VLP vaccine preparation to…"

We agree with the reviewers and the title has been revised as such “Epitope resurfacing on dengue virus-like particle vaccine preparation to induce broad neutralizing antibody”.

2) A clear acknowledgement of the heterogeneity from the outset, in the abstract and beyond. E.g. in abstract "herein we show for the first time that mD2VLP particles possess a T=1 icosahedral symmetry with a groove located within the E 52protein dimers near the 2-fold vertices that exposed highly overlapping, cryptic neutralizing 53 epitopes through cryo-electron microscopy reconstruction". This does not seem to be accurate. The mD2VLP preparation consist of entities that are mostly not this. The authors need to state specifically that the structure presented can guide vaccine design, but likely does not represent what is precisely in the vaccine.

We agree with the reviewers and the Abstract has been revised to acknowledge the size heterogeneity of dengue VLP vaccine preparation.

Also, mD2VLP with diameter of 31nm as epitope-resurfacing features has been further emphasized in the Results section and the Discussion section.